# SAFT: Structure-Aware Fine-Tuning of Large Language Models for AMR-to-Text Generation

## Abstract

Large Language Models (LLMs) are increasingly applied to tasks involving structured inputs such as semantic graphs, yet adapting them to such inputs remains non-trivial. Common approaches either linearize graphs, discarding structural information, or rely on specialized architectures that are not directly compatible with standard pretrained LLMs. We present SAFT, a structure-aware fine-tuning method that augments LLMs with graph positional encodings derived from the magnetic Laplacian of the input graph. These encodings are projected into the LLM embedding space, introducing relational inductive bias without modifying the model architecture. While SAFT is conceptually applicable to any task involving directed graph inputs with node–token alignment, we focus on the task of generating natural language text from an input AMR (Abstract Meaning Representation) graph, a directed graph encoding predicate-argument semantics of natural language sentences. AMR-to-text generation requires models to integrate both linguistic fluency and structural faithfulness, making it a demanding evaluation setting. We show that SAFT consistently improves or matches standard fine-tuning across all tested model families and scales, with gains that increase with graph structural complexity, both on sentence-level graphs of increasing depth and on document-level graphs of increasing size, demonstrating that structural encoding provides a reliable and scalable inductive bias for LLM fine-tuning.

## 1 Introduction

Large Language Models (LLMs) have become the dominant paradigm for natural language processing, demonstrating strong generalization across a wide range of sequential tasks. Recent work has sought to extend the reasoning and representation capabilities of LLMs beyond sequential data to more expressive, structured modalities, such as graphs (Jin et al., 2024a; Jiang et al., 2023; Fatemi et al., 2024; Zhang et al., 2022; Tang et al., 2024), motivated by the need for models that can reason over relational data. Yet, common approaches either require architectural modifications, or auxiliary components which limit their compatibility with pretrained general-purpose sequence models that characterize the recent LLM paradigm.

A particularly well-defined and linguistically grounded graph representation is the Abstract Meaning Representation (AMR) (Banarescu et al., 2013), i.e., a rooted, directed acyclic graph that encodes predicate-argument structure and core semantic relations. We focus on the AMR-to-text generation task, i.e., producing a natural language sentence that accurately expresses the meaning of an AMR graph. This task represents a strong benchmark for evaluating the ability of LLMs to interface with structured semantic representations, as it demands sensitivity to graph topology and semantic content while preserving fluency and coherence in the generated output.

Despite its importance, AMR-to-text generation remains challenging due to the inherent relational and semantic structure of AMRs. Prior approaches fall into three categories: sequence-to-sequence models (Bevilacqua et al., 2021; Cheng et al., 2022) that linearize AMRs and discard structural topology; graph-to-sequence methods (Song et al., 2018; Zhu et al., 2019; Ribeiro et al., 2021) that use task-specific encoders (e.g., Graph Neural Networks (GNNs)), which are not directly compatible with pretrained LLMs; and LLM

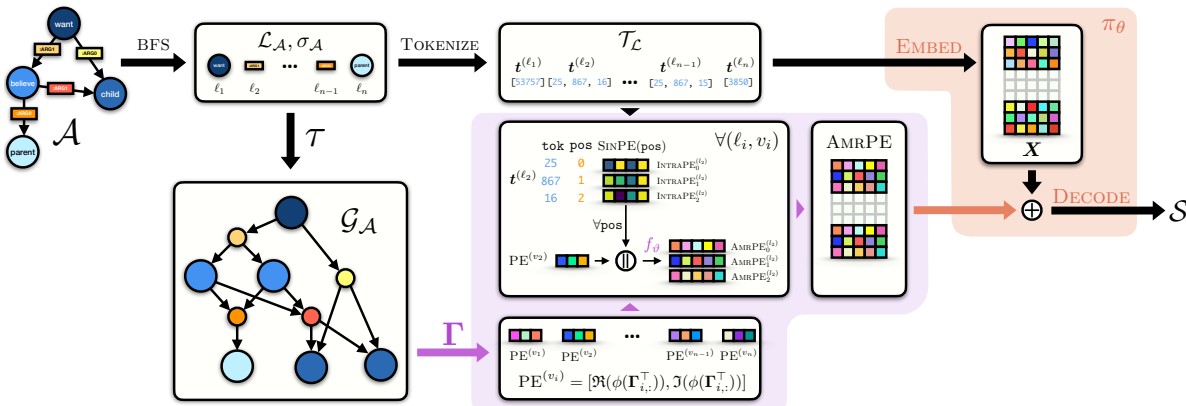

Figure 1: **Overview of SAFT.** An AMR graph $\mathcal{A}$ is first linearized into a token sequence $\mathcal{L}_\mathcal{A}$. We then construct a graph transformation $\mathcal{G}_\mathcal{A}$ and compute structure-aware positional encodings from its magnetic Laplacian. These encodings are combined with standard token positions to form AMR-specific embeddings (AMRPE). A simple MLP $f_\vartheta$ aligns them with the embedding space of the LLM $\pi_\theta$, after which they are injected into the token embeddings $X$. The model is fine-tuned to generate text $\mathcal{S}$, enabling structure-aware AMR-to-text generation without altering the LLM architecture.

adaptations (Mager et al., 2020; Yao et al., 2024a; Raut et al., 2025) that fine-tune or prompt-tune on linearized AMRs, ignoring the underlying graph structure.

The limitations of current approaches in terms of structure modeling and adaptation to pretrained sequence models, reveals a critical gap: *how can structural inductive bias be introduced into pretrained LLMs without architectural modification?*

We address this question with **SAFT**, a structure-aware fine-tuning method for LLMs that augments linearized graph inputs with positional encodings derived from the graph topology. Specifically, we compute graph positional encodings from the magnetic Laplacian (Furutani et al., 2020; Geisler et al., 2023) of an AMR-derived graph and inject them into the embeddings of the tokens of the graph linearization via a lightweight MLP. As shown in Figure 1, this injects relational inductive bias into any decoder-only LLM with minimal overhead.

SAFT demonstrates that simple, precomputed graph positional encodings, requiring only a directed graph and node–token alignment, can systematically improve LLM performance on structure-to-text tasks. We evaluate SAFT on AMR-to-text generation, a well-established structure-to-text task where inputs are directed acyclic graphs with rich relational edge semantics and evaluation is mature, making it a principled testbed for studying structure-aware generation with LLMs.

Our contributions include:

- A structure-aware fine-tuning framework for decoder-only LLMs that injects graph positional encodings derived from the magnetic Laplacian into token embeddings, introducing relational inductive bias without modifying the model architecture.

- A formulation of AMR-specific positional encodings that captures both graph directionality and structural topology via the eigenvectors of the magnetic Laplacian, combined with intra-node token positional encodings.

- Empirical evidence that SAFT consistently improves or matches standard fine-tuning across model families and scales (Table 1), with gains that increase with graph structural complexity both on sentence-level (Figure 2) and document-level inputs (Figure 3).

- An analysis of the representational geometry of AMRPEs, showing that adding structure-aware positional encodings expand the intrinsic dimensionality of the embedding space in directions largely orthogonal to pretrained token representations (Figure 4).

## 2 Background

We introduce graph representations and positional encodings, Abstract Meaning Representations (AMRs), and LLMs on structured inputs.

### 2.1 Graph Representations and Graph Positional Encodings

**Edge-labeled directed graphs.** AMRs are a prime example of edge-labeled directed graphs. We formally define these as tuples $\mathcal{A} = (\mathcal{V}_{\mathcal{A}}, \mathcal{E}_{\mathcal{A}}, \mathcal{R}_{\mathcal{A}}) \in \mathbb{A}$, where $\mathcal{V}_{\mathcal{A}}$ is the set of $n_{\mathcal{A}} = |\mathcal{V}_{\mathcal{A}}|$ nodes, $\mathcal{E}_{\mathcal{A}} \subseteq \mathcal{V}_{\mathcal{A}} \times \mathcal{V}_{\mathcal{A}}$ is the set of $m_{\mathcal{A}} = |\mathcal{E}_{\mathcal{A}}|$ directed edges, and $\mathcal{R}_{\mathcal{A}}$ is a finite set of relation types (edge labels), such that each edge $(u, v) \in \mathcal{E}_{\mathcal{A}}$ is associated with a label $r_{u,v} \in \mathcal{R}_{\mathcal{A}}$. The space of these graphs is denoted by $\mathbb{A}$.

**Edge-unlabeled directed graphs.** To understand positional encodings on graphs, it is useful to first consider edge-unlabeled directed graphs, defined as tuples $\mathcal{G} = (\mathcal{V}_{\mathcal{G}}, \mathcal{E}_{\mathcal{G}}) \in \mathbb{G}$, where $\mathcal{V}_{\mathcal{G}}$ is the set of $n_{\mathcal{G}} = |\mathcal{V}_{\mathcal{G}}|$ nodes, and $\mathcal{E}_{\mathcal{G}} \subseteq \mathcal{V}_{\mathcal{G}} \times \mathcal{V}_{\mathcal{G}}$ is the set of $m_{\mathcal{G}} = |\mathcal{E}_{\mathcal{G}}|$ directed, binary, and unlabeled edges, with $e_{u,v} = 1$ if there is a directed edge from node $u$ to node $v$, and 0 otherwise. The space of such graphs is denoted by $\mathbb{G}$. The relational structure is encoded in the adjacency matrix $\boldsymbol{A} \in \{0,1\}^{n_{\mathcal{G}} \times n_{\mathcal{G}}}$, where $\boldsymbol{A}_{u,v} = e_{u,v}$. We define the out-degree matrix $\boldsymbol{D}$ as a diagonal matrix with $\boldsymbol{D}_{u,u} = \sum_v \boldsymbol{A}_{u,v}$. Symmetrizing the adjacency matrix as $\boldsymbol{A}_S = \boldsymbol{A} \vee \boldsymbol{A}^\top$ (element-wise logical OR) yields an undirected representation of the graph, with a corresponding symmetrized degree matrix $\boldsymbol{D}_S$. While this is not *always* necessary, it is required in our approach, as it allows for the computation of the symmetric normalized Laplacian $\boldsymbol{L}_S = \boldsymbol{I} - \boldsymbol{D}_S^{-1/2} \boldsymbol{A}_S \boldsymbol{D}_S^{-1/2}$, which has a real eigendecomposition $\boldsymbol{L}_S = \boldsymbol{U} \boldsymbol{\Lambda} \boldsymbol{U}^\top$, where $\boldsymbol{U} \in \mathbb{R}^{n_{\mathcal{G}} \times n_{\mathcal{G}}}$ contains orthonormal eigenvectors and $\boldsymbol{\Lambda} = \mathrm{diag}(\lambda_1, \ldots, \lambda_{n_{\mathcal{G}}}) \in \mathbb{R}^{n_{\mathcal{G}} \times n_{\mathcal{G}}}$ is a diagonal matrix of real eigenvalues. However, this symmetrization inherently loses the directional information present in the original directed graph.

**Magnetic Laplacian.** To address the loss of directionality, we can employ the magnetic Laplacian (Furutani et al., 2020), which introduces directional information via complex-valued phase shifts. For $q \in \mathbb{R}_{\geq 0}$, the magnetic Laplacian $\boldsymbol{L}^{(q)} \in \mathbb{C}^{n_{\mathcal{G}} \times n_{\mathcal{G}}}$ is defined as:

$$\boldsymbol{L}^{(q)} := \boldsymbol{D}_S - \boldsymbol{A}_S \odot \exp\left(i \boldsymbol{\Theta}^{(q)}\right), \tag{1}$$

where $i = \sqrt{-1}$, $\odot$ denotes the Hadamard product (element-wise multiplication), and the phase matrix $\boldsymbol{\Theta}^{(q)} \in \mathbb{R}^{n_{\mathcal{G}} \times n_{\mathcal{G}}}$ is given by $(\boldsymbol{\Theta}^{(q)})_{u,v} = 2\pi q((\boldsymbol{A})_{u,v} - (\boldsymbol{A})_{v,u})$. The magnetic Laplacian $\boldsymbol{L}^{(q)}$ is Hermitian, guaranteeing a complete set of complex eigenvectors $\boldsymbol{\Gamma} \in \mathbb{C}^{n_{\mathcal{G}} \times n_{\mathcal{G}}}$.

**Graph Positional Encodings (GPEs).** GPEs assign each node a notion of position within the graph. Most common approaches leverage the spectral properties of the graph Laplacian to encode the structural position of nodes. In particular, the eigenvectors of the Laplacian provide an orthonormal basis that captures the graph's structure at varying frequencies.

Following (Belkin & Niyogi, 2003; Dwivedi & Bresson, 2021), we can define the Laplacian-based positional encoding $\phi(v_i) \in \mathbb{R}^k$ for a node $v_i \in \mathcal{V}_{\mathcal{G}}$ as the $i$-th row of the first $k$ eigenvectors in $\boldsymbol{U}$:

$$\phi(v_i) = [\boldsymbol{U}_{i,1}, \ldots, \boldsymbol{U}_{i,k}]^\top, \tag{2}$$

where $k \ll n_{\mathcal{G}}$ is a chosen dimensionality. These embeddings inherently capture coarse-to-fine structural patterns and are invariant to node permutations.

### 2.2 Abstract Meaning Representation

Abstract Meaning Representation (AMR) (Langkilde & Knight, 1998; Banarescu et al., 2013; Mansouri, 2025) is a semantic formalism that represents the meaning of a sentence as a rooted, directed acyclic graph

$\mathcal{A} = (\mathcal{V}_\mathcal{A}, \mathcal{E}_\mathcal{A}, \mathcal{R}_\mathcal{A})$. The nodes $\mathcal{V}_\mathcal{A}$ represent concepts, which are typically predicates, entities, or abstract ideas. The directed edges $\mathcal{E}_\mathcal{A} \subseteq \mathcal{V}_\mathcal{A} \times \mathcal{V}_\mathcal{A}$ capture the semantic relationships between these concepts. Each edge $(u, v) \in \mathcal{E}_\mathcal{A}$ is associated with a label $r_{u,v} \in \mathcal{R}_\mathcal{A}$, where $\mathcal{R}_\mathcal{A}$ is a finite set of predefined semantic roles. Common relation labels include `:ARG0` (agent), `:ARG1` (patient/theme), and `:mod` (modifier). A key characteristic of AMR is its abstraction from surface syntax, ensuring that sentences with equivalent semantics are mapped to isomorphic AMR graphs. For instance, the sentences "*The child wants the parent to believe them*" and "*What the child wanted is for the parent to believe them*" share the same underlying AMR structure.

**AMR Linearizations.** To enable the processing of AMR graphs by sequence-to-sequence models, such as LLMs, it is necessary to linearize the graph structure into a sequential format. This process, termed *linearization*, transforms an AMR graph $\mathcal{A}$ into a sequence of labels $\mathcal{L}_\mathcal{A} = (\ell_1, \ell_2, \ldots, \ell_L) \in \Sigma^*$, where each $\ell_i$ is a label from a predefined vocabulary $\Sigma$. A common serialization is the Penman notation (Kasper, 1989; Bateman, 1990; Goodman, 2020), a parenthetical representation that encodes the graph's concepts and relations in a compact textual form.

More recently, methods like breadth-first search (BFS) and depth-first search (DFS) based linearizations have been extended to AMRs (Bevilacqua et al., 2021). For example, BFS linearization traverses the graph level by level, employing special tokens to denote relation types and reentrancies, resulting in a structured sequence that aims at preserving the graph's information for autoregressive learning (Konstas et al., 2017). See Appendix A for notation details and visualizations.

## 2.3 Blueprint of Large Language Models

Large Language Models (LLMs) (Vaswani et al., 2017; Brown et al., 2020; Touvron et al., 2023) are parameterized functions $\pi_\theta : \Lambda^* \to \Delta^{|\Lambda|^m - 1}$ mapping an input token sequence $x \in \Lambda^*$ to a probability distribution over output sequences $y \in \Lambda^m$ of length $m$. Here, $\theta$ denotes the model parameters, $\Lambda$ the token vocabulary, and $\Delta^{|\Lambda|^m - 1}$ the probability simplex over $\mathbb{R}^{|\Lambda|^m}$. Outputs are generated autoregressively via $\pi_\theta(y \mid x) = \prod_{t=1}^m \pi_\theta(y_t \mid y_{<t}, x)$, with $y_{<t} = (y_1, \ldots, y_{t-1})$.

LLMs are typically pretrained on massive text corpora by predicting the next token in a sequence. This enables them to learn intricate linguistic and semantic patterns, resulting in strong generalization capabilities across various natural language processing tasks, often without task-specific supervision. To adapt a pretrained LLM to a specific downstream task, its parameters $\theta$ are fine-tuned on a task-specific dataset $\mathcal{D} = \{(x^{(i)}, y^{(i)})\}_{i=1}^N$, where $x^{(i)} \in \Lambda^*$ and $y^{(i)} \in \Lambda^m$.

# 3 Structure-Aware Fine-Tuning for AMR-to-Text Generation

We present SAFT, a lightweight method to incorporate graph topology into LLM fine-tuning without changing model parameters or architecture. The key idea is to inject graph positional encodings, derived from the magnetic Laplacian of the AMR graph, into the token embeddings during fine-tuning. This guides the model to better capture graph topology and long-range dependencies. Given an AMR graph $\mathcal{A} \in \mathbb{A}$, the goal is to generate a natural language sentence $\mathcal{S} \in \Sigma^*$ such that $\mathcal{S} = \psi(\mathcal{A})$ is fluent and semantically faithful to the input. We first apply a semantics-preserving transformation to the AMR graph $\mathcal{A}$ that enables spectral analysis (Section 3.1). From the magnetic Laplacian of $\mathcal{A}$, we extract node-level positional encodings, which combine with intra-node sinusoidal encodings to form token-level AMRPEs (Section 3.2). These are projected into the LLM embedding space and added to standard token embeddings during fine-tuning (Section 3.3). Figure 1 illustrates the pipeline; Algorithm 1 provides pseudocode.

## 3.1 Semantically-Preserving Transformation of AMR Graphs

Edge-labeled graphs, such as AMRs, pose a challenge for computing eigenvectors of the graph Laplacian, a standard step in deriving graph positional encodings. Applying the Laplacian directly would necessitate ignoring the crucial semantic information encoded in their edge labels. To overcome this, we introduce a transformation $\tau$ that converts a linearized AMR into a directed, edge-unlabeled *semantic-preserving graph* (SPG). The SPG retains the core semantics of the original AMR structure while enabling the application

of Laplacian-based spectral methods for positional encoding. This is similar in spirit to the reification used in Semantic Role Labeling or Resource Description Framework graphs (Marcheggiani & Titov, 2017; Marcheggiani & Perez-Beltrachini, 2018), but our transformation is tailored to AMRs and magnetic-Laplacian PEs.

**BFS Linearization.** We begin by applying a breadth-first search (BFS) traversal of the AMR graph $\mathcal{A}$, yielding a linearized sequence of labels $\mathcal{L}_{\mathcal{A}} = (\ell_1, \ldots, \ell_L) \in \Sigma^*$, where each $\ell_i$ represents a concept or role label. Each label corresponds to a node in the SPG $\mathcal{G}_{\mathcal{A}} = (\mathcal{V}_{\mathcal{G}}, \mathcal{E}_{\mathcal{G}})$, through an injective function $\sigma_{\mathcal{A}} : \mathbb{Z} \rightarrowtail \mathcal{V}_{\mathcal{G}}$ that aligns labels to their corresponding node in the graphs, i.e., $\sigma_{\mathcal{A}}(i) = v_i$. This implies $|\mathcal{V}_{\mathcal{G}}| = L$.

**SPG Transformation.** The transformation $\tau : \Sigma^* \times (\Sigma \rightarrowtail \mathcal{V}_{\mathcal{G}}) \to \mathbb{G}$ constructs the SPG $\mathcal{G}_{\mathcal{A}} = \tau(\mathcal{L}_{\mathcal{A}}, \sigma_{\mathcal{A}})$ from the label sequence and alignment mapping. Labeled edges in the original AMR are represented as role nodes in the SPG, preserving role semantics via directed edges to their source and target concept nodes. We unite co-referring nodes (e.g., marked with `<P1>`), and merge their connectivity. The resulting SPG is semantically equivalent to the original AMR but uses unlabeled edges for spectral compatibility, and makes re-entrancies and coreferences explicit. Additional details and visualization of the semantically-preserving transformation are available in Appendix B.1.

**Tokenization of Node Labels.** Each textual label $\ell_i \in \mathcal{L}_{\mathcal{A}}$ is associated with a node $v_i \in \mathcal{V}_{\mathcal{G}}$, $v_i = \sigma_{\mathcal{A}}(i)$. We tokenize each node label into a sequence of tokens $\boldsymbol{t}^{(\ell_i)} \in \Lambda^{p_i}$

$$\boldsymbol{t}^{(\ell_i)} = \text{TOKENIZE}(\ell_i) = (t_1^{(\ell_i)}, \ldots, t_{p_i}^{(\ell_i)}), \tag{3}$$

where $\Lambda$ denotes the tokenizer's vocabulary and $p_i$ is the number of tokens produced from the label $\ell_i$. When $p_i > 1$, we refer to $v_i = \sigma_{\mathcal{A}}(i)$ as a *multi-token node*.

## 3.2 AMR-Specific Positional Encodings

In the previous section we defined the transformation from an AMR graph $\mathcal{A}$ to its semantically-preserving representation $\mathcal{G}_{\mathcal{A}}$ that allows us to apply spectral graph theory and compute graph positional encodings. Here, we present our AMR-specific graph positional encodings, which we compute from the magnetic Laplacian (Section 2.1) of the semantic-preserving graph $\mathcal{G}_{\mathcal{A}}$. These encodings are meant to capture the topology of the AMR structure and its directionality.

**Node-level PEs.** The magnetic Laplacian, defined in Eq. (1), encodes directionality through complex phase shifts. We compute the magnetic Laplacian $\boldsymbol{L}^{(q)}$ of $\mathcal{G}_{\mathcal{A}}$ and extract the eigenvectors corresponding to the lowest $k$ eigenvalues, forming a complex matrix $\boldsymbol{\Gamma} \in \mathbb{C}^{n_{\mathcal{G}} \times k}$. Each node $v_i \in \mathcal{V}_{\mathcal{G}}$ is assigned a complex-valued $k$-dimensional embedding $\phi(v_i) \in \mathbb{C}^k$, $\phi(v_i) = [\boldsymbol{\Gamma}_{i,1}, \ldots, \boldsymbol{\Gamma}_{i,k}]^{\top}$. We convert $\phi(v_i)$ to a real-valued vector $\text{PE}^{(v_i)} \in \mathbb{R}^{2k}$ by concatenating the real and imaginary parts:

$$\text{PE}^{(v_i)} = [\Re(\phi(v_i)) \; \Im(\phi(v_i))]. \tag{4}$$

These positional encodings provide a spectral representation that reflects both local and global graph structure. Nodes with similar structural roles in the AMR, such as arguments or modifiers, will have similar embeddings, even if distant in the graph.

**Intra-node Token Positional Encodings.** For each token $t_j^{(\ell_i)}$ in $v_i = \sigma_{\mathcal{A}}(i)$ we apply sinusoidal positional encodings (Vaswani et al., 2017) to preserve their intra-node ordering:

$$\text{INTRAPE}_j^{(\ell_i)} = \text{SINPE}(j), \quad \text{for } j = 1, \ldots, p_i, \tag{5}$$

where $\text{INTRAPE}_j^{(\ell_i)} \in \mathbb{R}^d$. For single-token nodes ($p_i = 1$), we use $\text{INTRAPE}_1^{(\ell_i)} = \text{SINPE}(0)$.

**AMR Positional Encodings.** We combine the node-level and intra-node positional encodings to obtain the final AMR-specific positional encoding for each token $t_j^{(\ell_i)}$:

$$\text{AMRPE}_j^{(\ell_i)} = f_\vartheta \left( \text{PE}^{(v_i)} \,\|\, \text{IntraPE}_j^{(\ell_i)} \right), \tag{6}$$

where $\text{AMRPE}_j^{(\ell_i)} \in \mathbb{R}^{d_{\text{emb}}}$, $f_\vartheta : \mathbb{R}^{2k+d} \to \mathbb{R}^{d_{\text{emb}}}$ is a two-layer MLP with GeLU activation function (Hendrycks & Gimpel, 2016), and $\|$ denotes vector concatenation. This projection defines token-level positional encodings that captures (*i*) structural knowledge from the node-level positional encodings, and (*ii*) label-level sequential information from the intra-node token positional encodings. The embedding is mapped into the LLM embedding space ($d_{\text{emb}}$, see Section 3.3), allowing seamless integration during fine-tuning. Concatenating the positional encodings across all nodes/labels in their linearized order, as defined by $\sigma_\mathcal{A}$, determines the final AMR-specific positional encodings matrix:

$$\text{AMRPE} = \left( \text{AMRPE}_1^{(\ell_1)} \ldots \text{AMRPE}_{p_1}^{(\ell_1)} \ldots \text{AMRPE}_1^{(\ell_2)} \ldots \text{AMRPE}_{p_L}^{(\ell_L)} \right)^\top, \tag{7}$$

where $\text{AMRPE} \in \mathbb{R}^{p \times d_{\text{emb}}}$ and $p = \sum_{i=1}^L p_i$ is the total number of tokens in the linearization. $\text{AMRPE}$ is a representation of each token in the linearization that considers both the position of the token within its node-label and the global position in the graph.

## 3.3 LLM Fine-Tuning with AMR-Specific Positional Encodings

For ease of exposition, we represent the pretrained LLM decoder model as a composition:

$$\pi_\theta = \text{Decode} \circ \text{Embed}$$

where $\text{Embed} : \Lambda^p \to \mathbb{R}^{p \times d_{\text{emb}}}$ maps a sequence of tokens into the LLM's embedding space, and $\text{Decode} : \mathbb{R}^{p \times d_{\text{emb}}} \to \Sigma^*$ generates the output text sequence. Given an AMR graph $\mathcal{A}$, we obtain a linearized sequence of node labels $\mathcal{L}_\mathcal{A} = (\ell_1, \ldots, \ell_L)$ and their corresponding tokenized forms $\boldsymbol{t}^{(\ell_i)} = (t_1^{(i)}, \ldots, t_{p_i}^{(i)})$. The overall token sequence $\mathcal{T}_\mathcal{L} \in \Lambda^p$ is:

$$\mathcal{T}_\mathcal{L} = \boldsymbol{t}^{(\ell_1)} \,\|\, \boldsymbol{t}^{(\ell_2)} \,\|\, \ldots \,\|\, \boldsymbol{t}^{(\ell_L)} = (t_1, \ldots, t_p) \tag{8}$$

where $p = \sum_{i=1}^L p_i$ is the total number of tokens in the linearized graph. The sequence $\mathcal{T}_\mathcal{L}$ is mapped to the LLM embedding space as:

$$\boldsymbol{X} = \text{Embed}(\mathcal{T}_\mathcal{L}) \in \mathbb{R}^{p \times d_{\text{emb}}}. \tag{9}$$

**Integrating AMR positional encodings.** We then integrate structure-aware positional encodings to the embedded representation of the linearized sequence of tokens through the additive representation

$$\boldsymbol{H} = \boldsymbol{X} + \text{AMRPE} \in \mathbb{R}^{p \times d_{\text{emb}}}, \tag{10}$$

which preserves the dimensionality of the LLM embedding space. This modification affects only the input embeddings; the base model's positional encoding mechanism (e.g., RoPE (Su et al., 2024)) remains unchanged and continues to operate inside the attention layers. Finally, $\boldsymbol{H}$ is then fed to the LLM decoder components, including the transformer layers, head, and tokenizer, to return the generated output sequence $\mathcal{S} \in \Sigma^*$, as $\mathcal{S} = \text{Decode}(\boldsymbol{H})$.

**Prompt Handling.** For clarity and modularity, we exclude the prompt segment from the structure-aware positional encoding process. The prompt is tokenized independently from the AMR linearization to avoid disrupting the alignment between graph nodes and tokens. Its tokens are embedded using the standard learned embeddings without any additional positional encodings beyond those already handled by the pretrained model. This design simplifies the architecture and ensures that the inductive bias introduced by our positional encodings applies exclusively to the AMR portion of the input. Additional details are provided in Appendix B.

**Computational Considerations.** SAFT introduces negligible computational overhead relative to standard LoRA fine-tuning. All graph-related operations, such as BFS linearization, SPG construction, magnetic Laplacian computation, and eigendecomposition, are performed once per graph at preprocessing time and cached, adding on average 0.01 seconds per sample on AMR 3.0 (Table 10). The only runtime addition during training and inference is the forward pass through the MLP $f_\vartheta$, whose parameter count is negligible relative to the underlying LLM. Further details are provided in Appendix C.5.

# 4 Experiments

We evaluate our structure-aware fine-tuning approach on the AMR 3.0 benchmark. First, we show that SAFT achieves a new state of the art on sentence-level AMR-to-text generation (Section 4.2). We then demonstrate that it typically improves over conventional fine-tuning, with gains that become more pronounced at increased structural complexity (Section 4.3). We further extend this analysis to document-level AMRs, where SAFT yields even larger improvements (Section 4.4). To separate the effect of fine-tuning from potential pretraining exposure, we also evaluate state-of-the-art closed models in zero-shot and few-shot settings, showing that strong general-purpose LLMs still struggle with AMR-to-text generation without task-specific supervision (Section 4.5). Finally, we analyze how injecting AMRPEs affects the geometry of token representations in the model's latent space (Section 4.6).

## 4.1 Experimental Setup

We use AMR 3.0 (LDC2020T02) (Knight et al., 2020) and DocAMR, which incorporates discourse structure and inter-sentence dependencies; split details are provided in Appendix E. We fine-tune pretrained LLMs, including LLaMA 3.2 (1B, 3B) (Touvron et al., 2023), Qwen 2.5 (0.5B, 1.5B, 3B) (Bai et al., 2023), and Gemma (2B) (Team et al., 2024), with Low-Rank Adaptation (LoRA)[1] (Hu et al., 2022), both with our graph-based positional encoding module (SAFT) and without it (FT). Training and inference settings are identical across conditions. All native components of the base models are preserved; for example, rotary positional embeddings (RoPE) remain active[2]. We report BLEU (Papineni et al., 2002) and chrF (Popović, 2015) scores using greedy decoding, and additionally BERTScore (Zhang et al., 2020) and METEOR (Lavie & Agarwal, 2007) in Appendix C.4. All experiments are implemented in `LitGPT`; further details are in Appendix B.3.

## 4.2 AMR 3.0 results

We evaluate SAFT on AMR 3.0 against widely used AMR-to-text baselines: SPRING (Bevilacqua et al., 2021), StructAdapt (Ribeiro et al., 2021), and BiBL (Cheng et al., 2022). Unless explicitly noted, all models are trained on the same AMR 3.0 training split and evaluated on the official held-out test set. SPRING and BiBL additionally report variants trained with extra heuristically labeled data, which we gray out in Table 1. Both models rely on linearized AMR inputs and sequence-to-sequence training. StructAdapt, in contrast, introduces a dedicated graph encoder within an encoder–decoder architecture. Our approach differs in that it operates on modern decoder-only LLMs and injects structural information through graph positional encodings without introducing a separate encoder component.

The results in Table 1 present a comparative evaluation of prior approaches[3] (Bevilacqua et al., 2021; Cheng et al., 2022; Ribeiro et al., 2021), alongside our fine-tuned decoder-only LLMs, both with and without the proposed graph positional encoding module (SAFT). While these architectures are not directly matched, this comparison provides useful context for situating decoder-only models within existing AMR-to-text work. For baseline models, we include results for versions trained with and without extra heuristically labeled data where available. Notably, we find that fine-tuning several LLMs using LoRA (Hu et al., 2022) already yields improvements over earlier models, including those that use extra training data. Our proposed approach, SAFT, further boosts performance, highlighting the benefit of incorporating structural information in the

---

[1]SAFT is compatible with any parameter-efficient tuning method.
[2]AMRPEs do not alter or replace the LLM's own positional encoding.
[3]Reported scores are taken directly from the original publications and not reproduced.

form of graph positional encodings during LLM fine-tuning. As shown in Table 1, SAFT consistently improves or matches FT, with aggregate BLEU gains of +0.8 over the FT variant across the different models. However, such aggregate scores can obscure important variation across inputs of different structural complexity. To more accurately characterize when and how SAFT helps, we turn to a stratified analysis.

### 4.3 Complexity-stratified results

To evaluate performance variation with input complexity, we stratify the evaluation by AMR graph depth $\delta(\mathcal{A})$, measured on the original AMR $\mathcal{A}$ prior to preprocessing, by grouping examples where $\delta(\mathcal{A}) \geq z$ for varying thresholds $z$. We then calculate the BLEU score on these stratified subsets to quantify how our structure-aware fine-tuning approach improves performance at increasing $\delta(\mathcal{A})$ with respect to conventional fine-tuning. We define the following metric:

$$\Delta_{\text{BLEU}}(z) = \text{BLEU}_{\text{SAFT}}^{z} - \text{BLEU}_{\text{FT}}^{z}, \qquad (11)$$

where $\text{BLEU}_{\text{SAFT}}^{z}$ and $\text{BLEU}_{\text{FT}}^{z}$ represent the BLEU scores achieved by SAFT and conventional fine-tuning (FT), respectively, on instances with AMR depth $\delta(\mathcal{A}) \geq z$. Figure 2a shows the extent to which SAFT improves the performance of LLMs at increasing levels of semantic complexity. At the highest evaluated threshold ($\delta(\mathcal{A}) \geq 8$), SAFT surpasses FT by +1.1 to +4.4 BLEU across all model families.

The divergence of the curves at greater depths and an overall upward trend across all models highlight the increasing importance of modeling structure explicitly as semantic complexity grows. While Gemma 2B showed little overall improvement across the full dataset (see Table 1), the benefit of SAFT becomes more pronounced on examples with greater semantic complexity.

To further assess how the benefit varies with respect to depth-one AMRs (i.e., $\delta(\mathcal{A}) = 1$), we define a second-order delta which measures the change in improvement relative to these simple structures:

Table 1: BLEU and ChrF scores on the AMR 3.0 test set. We compare prior encoder-decoder systems (top) with decoder-only LLMs fine-tuned under standard fine-tuning (FT) or SAFT (bottom). Cross-group comparisons should be interpreted with caution due to differences in backbone architecture and pretraining scale. All models use AMR 3.0 training data only, unless marked with additional corpora (grayed out). Best results per metric among models without additional data are bolded; relative improvement of SAFT over FT is reported in brackets.

| Model | Variant | BLEU ↑ | CHRF ↑ |
|---|---|---|---|
| *Previous Work* | | | |
| BiBL | w/o Extra data | 47.4 | 74.5 |
| | + Extra data | 50.7 | 76.7 |
| SPRING | w/o Extra data | 44.9 | 72.9 |
| | + Extra data | 46.5 | 73.9 |
| StructAdapt | w/o Extra data | 48.0 | 73.2 |
| *Our Finetuned LLMs: trained without extra data* | | | |
| LLaMA 3.2 (3B) | FT | 53.5 | 75.5 |
| | SAFT | **54.2** (+1.3%) | **76.0** (+0.7%) |
| LLaMA 3.2 (1B) | FT | 45.5 | 70.9 |
| | SAFT | 47.8 (**+5.1%**) | 71.9 (+1.4%) |
| Qwen 2.5 (3B) | FT | 51.6 | 72.1 |
| | SAFT | 51.9 (+0.6%) | 74.8 (**+3.7%**) |
| Qwen 2.5 (1.5B) | FT | 50.5 | 73.7 |
| | SAFT | 51.7 (+2.4%) | 74.5 (+1.1%) |
| Qwen 2.5 (0.5B) | FT | 42.7 | 69.0 |
| | SAFT | 42.9 (+0.5%) | 69.3 (+0.4%) |
| Gemma (2B) | FT | 52.9 | 73.5 |
| | SAFT | 52.9 (+0.1%)* | 73.6 (+0.1%) |

*Not rounded scores: 52.87 (FT) vs. 52.91 (SAFT).

$$\Delta_{\text{BLEU}}^{1}(z) = \Delta_{\text{BLEU}}(z) - \Delta_{\text{BLEU}}(1). \qquad (12)$$

Figure 2b further validates the finding that SAFT delivers increasing gains on more complex AMRs. Specifically, it shows the relative improvement of each model compared to its own performance on depth-one AMRs. The positive upward trends across all models indicate that the advantage of incorporating graph structure grows with semantic complexity. This consistent behavior across model families and sizes reinforces the scalability and applicability of our method across different architectures and parameter scales.

### 4.4 DocAMR results

To assess how SAFT performs on structurally more complex inputs, we evaluate on DOCAMR, a document-level subset of AMR 3.0 (Knight et al., 2020). Its test set is constructed by merging sentence-level AMRs drawn directly from the AMR 3.0 test split into documents, augmented with inter-sentence coreference edges. Both FT and SAFT models are fine-tuned on the AMR 3.0 training split and are never exposed to

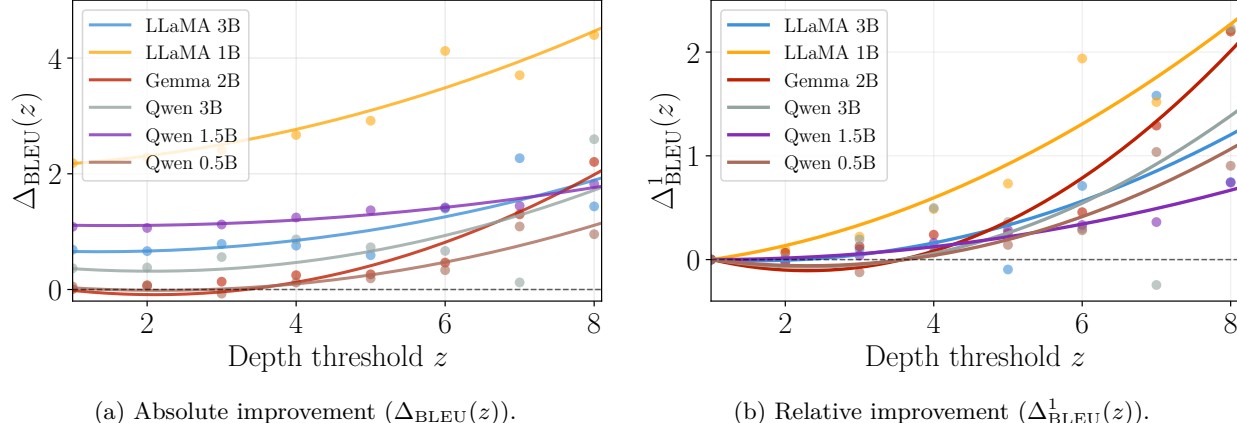

(a) Absolute improvement ($\Delta_{\text{BLEU}}(z)$).    (b) Relative improvement ($\Delta^1_{\text{BLEU}}(z)$).

Figure 2: BLEU score difference between SAFT and FT models, evaluated on the subset of AMR 3.0 test instances with graph depth $\delta(\mathcal{A}) \geq z$. Panel (a) shows absolute differences $\Delta_{\text{BLEU}}$; panel (b) shows differences normalized by each model's performance at depth-1 graphs $\Delta^1_{\text{BLEU}}$. Lines are second-degree polynomial fits. Both panels show a consistent increase in SAFT's advantage as structural complexity grows.

document-level structures during fine-tuning. The DocAMR evaluation therefore does not introduce new content or vocabulary: the underlying sentences are identical to those in the standard AMR 3.0 test set. What changes is the structural complexity: sentence-level graphs are composed into larger, more densely connected structures with cross-sentence dependencies. This setup isolates the effect of structural complexity from content shift, allowing us to directly assess whether the graph-based positional encodings introduced by SAFT transfer from simple sentence-level graphs to compositionally richer inputs without document-level supervision.

As shown in Figure 3, SAFT consistently improves performance across all levels of complexity which is measured by $\#_{\text{AMR}}$, the number of AMR graphs contained within a document-level AMR, indicating the overall size and structural density of the input. While the downward trend shows that all models struggle with deep topologies, the consistent, and occasionally increasing, improvement shows that using SAFT improves performance on more structurally-dense inputs. The performance gap between SAFT and conventional fine-tuning (FT) widens with increasing AMR complexity, suggesting that structural inductive bias becomes increasingly crucial in complex document-level generation. This reinforces our central claim that structure-aware fine-tuning is especially beneficial when models must reason over longer contexts and inter-sentential

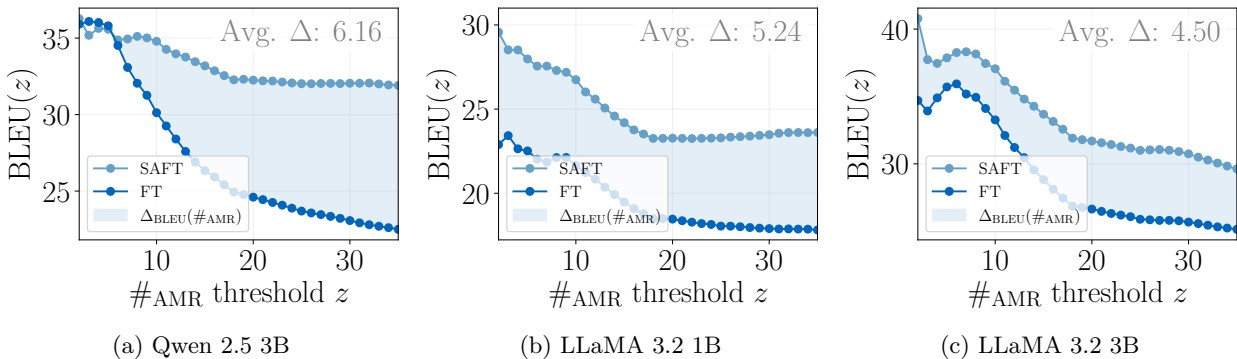

(a) Qwen 2.5 3B    (b) LLaMA 3.2 1B    (c) LLaMA 3.2 3B

Figure 3: BLEU scores of SAFT and FT models on the DocAMR test set, evaluated cumulatively on document-level AMRs containing at least $z$ sentence-level graphs. Both models were trained on sentence-level AMR 3.0 only. Average BLEU improvement of SAFT over FT across document sizes is reported per model.

relations. We excluded the Gemma model from this experiment due to its limited context window (4096 tokens), which could not accommodate DocAMR inputs.

**Results summary.** Conventional fine-tuning (FT) of LLMs already surpasses prior non-LLM baselines on AMR-to-text generation. Our structure-aware fine-tuning approach (SAFT), further improves performance across model families and scales. The improvements are especially consistent on semantically complex and document-level inputs, confirming that graph-based positional encodings enhance the model's ability to reason over AMR structure.

## 4.5 Evaluation under zero-shot and few-shot prompting on SoTA LLMs

We additionally evaluate two closed-source large language models, GPT-4o-mini and GPT-4o, in both zero-shot and few-shot settings. We use a fixed task prompt with seven AMR–text examples; the full template is provided in Appendix B.3.2. As shown in Table 2, moving from zero-shot to few-shot leads to only marginal improvements for both models. Despite their scale, their performance remains substantially below that of a much smaller 3B model fine-tuned on AMR. In fact, standard fine-tuning (FT) of Qwen 2.5 (3B) already more than doubles the BLEU score of GPT-4o-mini, and SAFT further improves on FT using the same backbone. This suggests that AMR-to-text generation is not easily solved through prompting alone under our prompting setup. In-context examples provide limited signal about the underlying graph structure, whereas fine-tuning exposes the model to the full supervision of the training set and allows SAFT to inject explicit structural information directly.

Table 2: BLEU, ChrF, METEOR, and BERTScore on the AMR 3.0 test set for closed-source models under zero-shot and few-shot prompting, compared to fine-tuned Qwen 2.5 3B. Few-shot prompting uses seven fixed AMR–text examples. Best results per metric are bolded.

| Model | Setting | BLEU ↑ | CHRF ↑ | METEOR ↑ | BERTScore ↑ |
|---|---|---|---|---|---|
| GPT-4o-mini | Zero-shot | 16.3 | 44.8 | 38.4 | 74.8 |
|  | Few-shot | 17.0 | 45.2 | 39.9 | 75.7 |
| GPT-4o | Zero-shot | 28.0 | 59.8 | 52.1 | 81.1 |
|  | Few-shot | 29.6 | 60.1 | 52.4 | 81.8 |
| Qwen 2.5 (3B) | FT | 51.6 | 72.1 | 59.4 | 82.9 |
|  | SAFT (ours) | **51.9** | **74.8** | **60.1** | **83.7** |

## 4.6 Geometric Analysis of AmrPEs

We analyze how AMRPEs modify the geometry of token embeddings $X$ at the point of injection, producing augmented representations $H = X + \text{AMRPE}$ (Eq. (10)). The goal is to characterize the structure of the injected signal itself; whether this injection translates into effective generation is addressed empirically in Section 4.2. Figure 4 summarizes our findings across four properties.

**The structural signal is consistently larger in norm (a).** The norm ratio $\|\text{AMRPE}\|_2/\|X\|_2$ is distributed around a median of approximately 3, with a right skew indicating that a non-trivial fraction of tokens receive an even stronger structural injection. AMRPEs therefore constitute a substantial additive signal rather than a small perturbation of the token embeddings.

**The injection is nearly orthogonal to the token embedding space (b).** The cosine similarity between AMRPE and $X$ is sharply concentrated near zero and holds consistently across the full dataset rather than only on average, indicating that the structural signal occupies directions largely linearly independent from the pretrained textual representations. This near-orthogonality means that $X$ is not directly overwritten at the point of injection. We note, however, that downstream operations such as layer normalization and attention are sensitive to magnitude, so the norm asymmetry established in panel (a) may still influence how subsequent transformer layers process $H$.

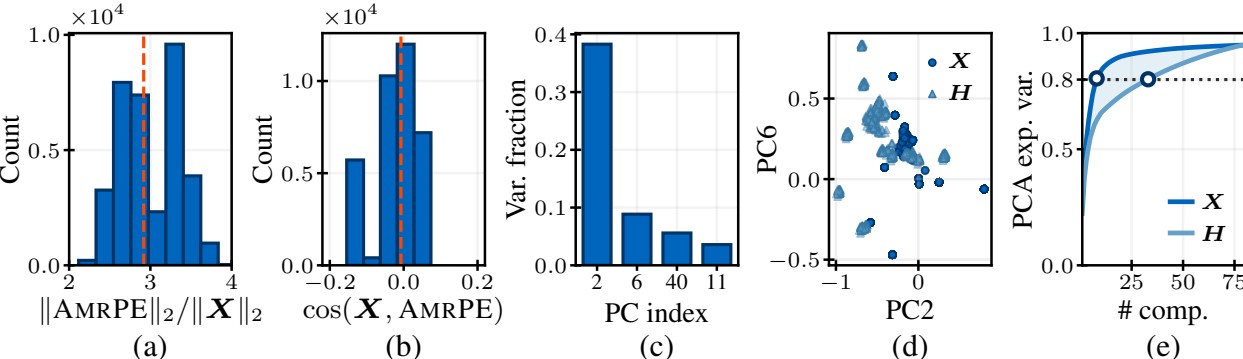

Figure 4: Geometric analysis of the effect of adding AMRPE to token embeddings $\boldsymbol{X}$, yielding $\boldsymbol{H} = \boldsymbol{X} + \text{AMRPE}$. (a) Distribution of the norm ratio $\|\text{AMRPE}\|_2/\|\boldsymbol{X}\|_2$. (b) Distribution of cosine similarity between AMRPE and $\boldsymbol{X}$. (c) Variance of AMRPE projected onto the PCA basis of $\boldsymbol{X}$, by principal component index. (d) Scatter of $\boldsymbol{X}$ and $\boldsymbol{H}$ in the top-2 PCA dimensions of $\boldsymbol{X}$. (e) Cumulative explained variance spectra of $\boldsymbol{X}$ and $\boldsymbol{H}$.

**The injected signal is structured and induces a coherent directional shift (c–d).** Projecting AMRPEs onto the PCA basis of $\boldsymbol{X}$ reveals that their variance is concentrated almost entirely on a single dominant principal component, with rapid decay across remaining axes. This anisotropy is not accidental: projecting both $\boldsymbol{X}$ and $\boldsymbol{H}$ into the same PCA space shows that the augmented representations undergo a coherent, directed shift along this dominant axis, with residual variance distributed across many weaker components. The structural signal is therefore highly organized rather than diffuse noise injected uniformly across the embedding space.

**The augmented space has higher intrinsic dimensionality (e).** The explained-variance spectrum of $\boldsymbol{H}$ is markedly flatter than that of $\boldsymbol{X}$: whereas $\boldsymbol{X}$ reaches 80% cumulative variance at approximately 10 principal components, $\boldsymbol{H}$ requires roughly 30. This expansion reflects the introduction of structured signal along directions not already present in the token embeddings, and is a direct consequence of the near-orthogonality established in panel (b).

Taken together, these four properties characterize AMRPEs as a large, orthogonal, and highly structured additive signal that expands the intrinsic dimensionality of the token representation space. The geometric picture is consistent with an injection mechanism that adds new representational directions rather than displacing existing ones, though orthogonality at the point of addition does not guarantee that the magnitude asymmetry is inconsequential for downstream transformer computations. The performance results in Section 4.2 provide the functional evidence that this injection is beneficial across all tested model families.

## 5 Related Work

To the best of our knowledge, SAFT is the first work to inject graph positional encodings into LLMs for structure-aware fine-tuning without modifying the model's architecture. SAFT is architecture-agnostic: it introduces relational inductive bias through precomputed, parameter-free encodings (from the magnetic Laplacian spectrum), injected via a lightweight projection into the LLM's embedding space. This eliminates the need for graph-specific training while allowing direct fine-tuning of the LLM. Prior work on graph-to-text generation, particularly AMR-to-text, can be grouped into three families: **linearization-based**, **adapter-based**, and **graph-tuning-based**. We discuss here some of the main approaches in each family and report additional details in Appendix F.

**Linearization-based.** Graphs are serialized into sequences for seq2seq models such as BART or T5. Examples include SPRING (Bevilacqua et al., 2021), AMR-BART (Bai et al., 2022), and BiBL (Cheng et al.,

2022), which differ in traversal strategies and auxiliary tasks. LLMs have also been adapted via fine-tuning (Raut et al., 2025; Mager et al., 2020) or prompting (Yao et al., 2024a; Jin et al., 2024b). Similar ideas extend beyond AMR to molecular graphs (Zheng et al., 2024), tables (Fang et al., 2024), and 3D meshes (Wang et al., 2024).

**Adapter-based.** These approaches introduce graph-native components, such as GNN-based adapters or modified attention layers, to inject relational information. StructAdapt (Ribeiro et al., 2021) uses graph-aware adapters within pretrained transformers, while others directly encode AMR graphs via graph-to-sequence models (Zhu et al., 2019; Song et al., 2018; Wang et al., 2020).

**Graph-tuning-based.** Recent methods integrate learned graph modules with LLMs, for example by training GNN-based adapters (Huang et al., 2024), adding graph transformers at each layer (Chai et al., 2023), or prepending GNN-derived embeddings into the prompt (Tang et al., 2024). While effective, these strategies require additional trainable modules, architectural changes, or costly pretraining.

## 6 Conclusion

We introduce SAFT, a structure-aware fine-tuning method that augments decoder-only LLMs with graph positional encodings derived from the magnetic Laplacian of AMR graphs, injected into the token embedding space without modifying the model architecture. Across six decoder-only LLMs ranging from 0.5B to 3B parameters, SAFT consistently improves over standard fine-tuning, with gains increasing consistently with graph structural complexity on both sentence-level AMR 3.0 and document-level DocAMR. Geometric analysis of the token representation space reveals that AMRPEs are injected along directions largely orthogonal to the pretrained token embeddings, inducing a consistent directional shift and increasing the intrinsic dimensionality of the representation space, which suggests a potential mechanism for structural enrichment without overwriting pretrained semantic content, though this remains to be directly verified. When applied to recent decoder-only LLMs, SAFT models surpass prior encoder-decoder systems trained on equivalent data, and remain competitive with those using additional fine-tuning corpora, despite the architectural differences between these paradigms. Whether these findings generalize beyond AMR-to-text generation to other graph-structured tasks, such as knowledge graph verbalization or discourse-to-text, remains an open empirical question we leave for future work.

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

## A  Abstract Meaning Representation

### A.1  Reference sentence

We report here the representations of the sentence used in the main body (Section 2.2) as reference:

*The child wants the parent to believe them.*

Figure 5a is the graph representation of the sentence, while Figures 5b and 5c are the Penman and BFS linearizations, respectively.

(a) Graph representation    (b) Penman notation    (c) BFS linearization

Figure 5: Three aligned representations of the sentence *"The child wants the parent to believe them."*: (a) a graph-based AMR structure, (b) its corresponding Penman notation, and (c) a BFS linearization used for sequence-based processing.

## A.2 Visualizing other AMR examples

We visualize the linearization and preprocessing steps, as detailed in Appendix B.1 for additional sentences. Table 3 summarizes the examples and points to the corresponding figures showing the original AMR graphs and the transformation pipeline.

Table 3: Examples with sentences, AMR graphs, and preprocessing steps

| Ex. | Sentence | Original AMR Graph | Preprocessing Steps |
|-----|----------|--------------------|--------------------|
| I   | I used to play tennis. | Figure 11 | Figure 12 |
| II  | This is really eye-opening. | Figure 13 | Figure 14 |
| III | The key is to be as objective as possible. | Figure 15 | Figure 16 |
| IV  | Speeding and accidents have surged as well. | Figure 17 | Figure 18 |

# B Implementation details

## B.1 Semantically-preserving AMR transformation

We report detailed information of our proposed semantically-preserving transformation $\tau : \Sigma^* \times (\Sigma \rightarrowtail \mathcal{V}_\mathcal{G}) \to \mathbb{G}$ of AMR graph (Section 3.1).

Given a linearization (label sequence) $\mathcal{L}_\mathcal{A}$ and alignment $\sigma_\mathcal{A}$—as discussed in Section 3.1 and shown in Figure 6b—the transformation $\tau$ constructs the SPG $\mathcal{G}_\mathcal{A} = \tau(\mathcal{L}_\mathcal{A}, \sigma_\mathcal{A})$ through the following steps:

1. **Substructure Construction** (TOSUBGRAPH, Figure 6c): $\mathcal{L}_\mathcal{A}$ is segmented at each `<stop>` token. Each segment defines a local subgraph rooted at a head concept and includes its outgoing role-labeled edges (e.g., `:ARG0`) and target nodes.

$$\{\bar{\mathcal{A}}_i\}_i = \text{TOSUBGRAPH}(\mathcal{L}_\mathcal{A}).$$

2. **Edge-to-Node Conversion** (ROLEEXPAND, Figure 6d): Each labeled edge $(u \xrightarrow{r} v)$ is expanded into a role node $r$, creating two unlabeled edges: $(u \to r)$ and $(r \to v)$. This yields a directed graph with no edge labels.

$$\bar{\mathcal{G}}_i = \text{ROLEEXPAND}(\bar{\mathcal{A}}_i).$$

3. **Stop Node Re-insertion** (ADDSTOPNODES, Figure 6d): The `<stop>` labels are inserted in each subgraph as a special terminal node. These nodes mark the end of node expansions and, alongside $\sigma_\mathcal{A}$, support alignment between graph nodes and tokens in $\mathcal{L}_\mathcal{A}$.

$$\hat{\mathcal{G}}_i = \text{ADDSTOPNODES}(\bar{\mathcal{G}}_i).$$

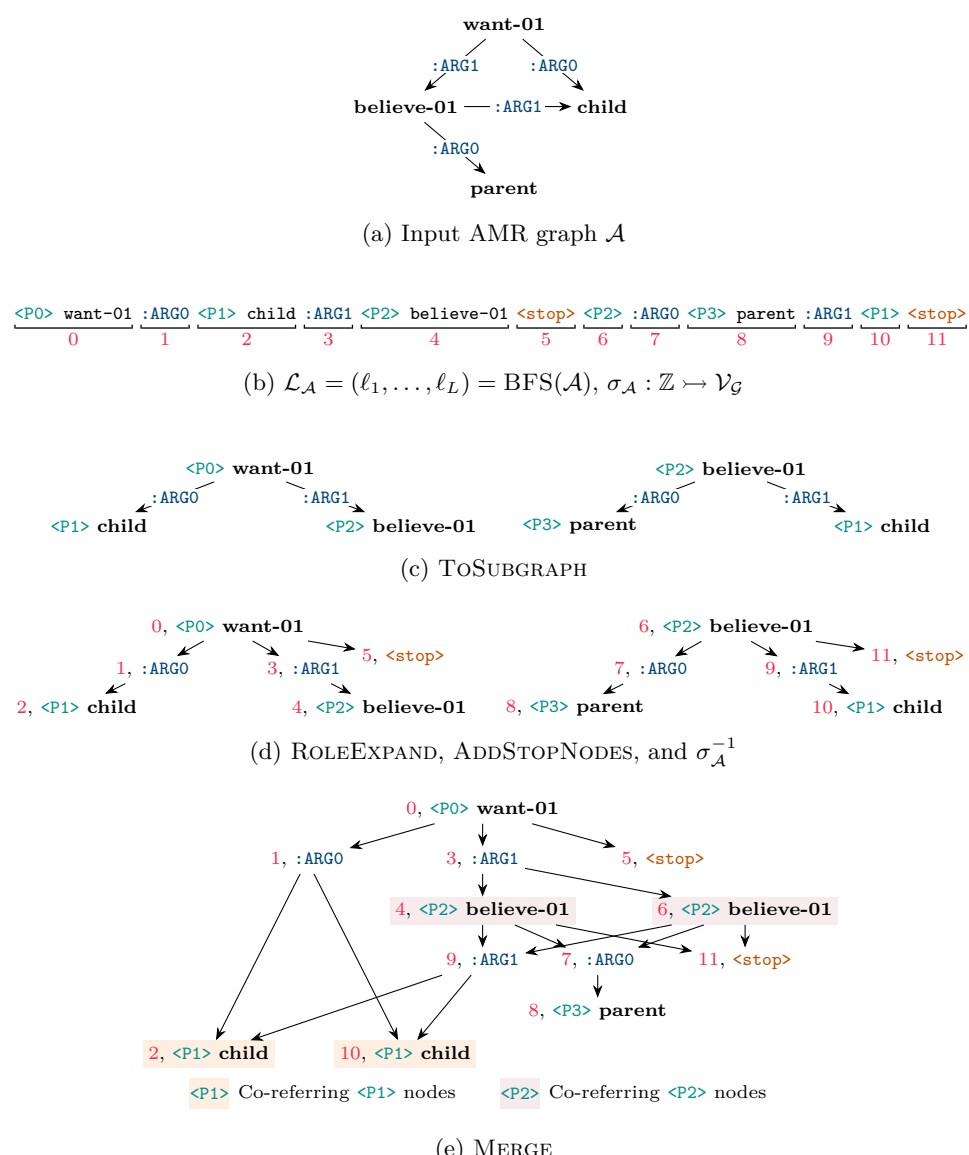

Figure 6: Transformation steps from an AMR graph $\mathcal{A}$ to the Role-Expanded Graph $\mathcal{G}$.

4. **Node Ordering Assignment** ($\sigma_{\mathcal{A}}^{-1}$, Figure 6d): Assign to each node an index inherited from the BFS order to preserve alignment between token positions in $\mathcal{L}_{\mathcal{A}}$ and graph nodes in $\mathcal{G}$.

$$i_v = \sigma_{\mathcal{A}}^{-1}(v), \quad \forall v \in \hat{\mathcal{G}}_i.$$

5. **Pointer-Based Merging** (MERGE, Figure 6e): For each pointer index $j$ (e.g., `<P2>`), identify co-referring nodes $\mathcal{U}_j = \{u_1, \ldots, u_k\}$ such that all $u_i$ share pointer $j$. Then:

   (a) Merge *incoming* edges: $\mathcal{E}_{\mathcal{U}_j}^{\text{in}} = \bigcup_{i=1}^{k} \mathcal{E}^{\text{in}}(u_i)$, with $\mathcal{E}^{\text{in}}(u_i) = \{(v, u_i) : (v, u_i) \in \mathcal{E}_{\mathcal{A}}\}$.

   (b) Merge *outgoing* edges: $\mathcal{E}_{\mathcal{U}_j}^{\text{out}} = \bigcup_{i=1}^{k} \mathcal{E}^{\text{out}}(u_i)$, with $\mathcal{E}^{\text{out}}(u_i) = \{(u_i, v) : (u_i, v) \in \mathcal{E}_{\mathcal{A}}\}$

   (c) Update the connectivity for each $u_i \in \mathcal{U}_j$: $\mathcal{E}^{\text{in}}(u_i) := \mathcal{E}_{\mathcal{U}_j}^{\text{in}}, \quad \mathcal{E}^{\text{out}}(u_i) := \mathcal{E}_{\mathcal{U}_j}^{\text{out}}$

$$\mathcal{G}_{\mathcal{A}} = \text{MERGE}(\{\bar{\mathcal{G}}_i\}_i).$$

## B.2 Algorithm

---

**Algorithm 1** AMR-to-Text Generation with SAFT

---

**Input:** AMR graph $\mathcal{A}$, pretrained LLM $\pi_\theta = \text{DECODE} \circ \text{EMBED}$
**Output:** Generated text sequence $\mathcal{S}$

1: $\mathcal{L}_\mathcal{A}, \sigma_\mathcal{A} = \text{BFS}(\mathcal{A})$                  ▷ Linearize AMR and align labels
2: $\mathcal{G}_\mathcal{A} = \tau(\mathcal{L}_\mathcal{A}, \sigma_\mathcal{A})$           ▷ Transform to semantic-preserving graph (SPG)
3: $\mathbf{\Gamma} = \text{MAGLAPEVD}(\mathcal{G}_\mathcal{A}, k)$        ▷ Compute magnetic Laplacian eigvecs
4: **for** each $(i, \ell_i) \in \text{enumerate}(\mathcal{L}_\mathcal{A})$ **do**
5:      $v_i = \sigma_\mathcal{A}(i)$
6:      $\boldsymbol{t}^{(\ell_i)} = \text{TOKENIZE}(\ell_i)$            ▷ Tokenize label (Eq. (3))
7:      $\phi(v_i) = \mathbf{\Gamma}_{i,:}^\top$            ▷ Select complex eigenvector
8:      $\text{PE}^{(v_i)} = [\Re(\phi(v_i)) \; \Im(\phi(v_i))]$      ▷ Node-level PE (Eq. (4))
9:      **for** each token $(j, t_j^{(\ell_i)}) \in \text{enumerate}(\boldsymbol{t}^{(\ell_i)})$ **do**
10:          $\text{INTRAPE}_j^{(\ell_i)} = \text{SINPE}(j)$      ▷ Intra-node PE (Eq. (5))
11:          $\text{AMRPE}_j^{(\ell_i)} = f_\theta(\text{PE}^{(v_i)} \| \text{INTRAPE}_j^{(\ell_i)})$      ▷ Token-wise AMR PE (Eq. (6))
12:      **end for**
13: **end for**
14: $\text{AMRPE} = (\text{AMRPE}_1^{(\ell_1)} \ldots \text{AMRPE}_{p_L}^{(\ell_L)})^\top$      ▷ AMR PE (Eq. (7))
15: $\mathcal{T}_\mathcal{L} = \boldsymbol{t}^{(\ell_1)} \| \boldsymbol{t}^{(\ell_2)} \| \ldots \| \boldsymbol{t}^{(\ell_L)}$      ▷ Token sequence (Eq. (8))
16: $\boldsymbol{X} = \text{EMBED}(\mathcal{T}_\mathcal{L})$      ▷ Token sequence embedding
17: $\boldsymbol{H} = \boldsymbol{X} + \text{AMRPE}$      ▷ Inject structure-aware PE (Eq. (10))
18: $\mathcal{S} = \text{DECODE}(\boldsymbol{H})$      ▷ Generate output sequence

---

## B.3 Training details

**Training setup.** We build on the open-source `LitGPT`[4] framework and extend it to incorporate our structure-aware representations. In particular, we (i) generate graph-based positional encodings (AMRPE), (ii) align AMR nodes to token positions via node-aware tokenization, (iii) introduce a lightweight projection layer $f_\theta$ to map these encodings into the LLM's embedding space, and (iv) add task-specific prompting to support AMR-to-text generation.

As in standard sequence-to-sequence fine-tuning, we optimize a cross-entropy objective. The pretrained model weights are updated only through LoRA adapters, while the parameters of $f_\theta$ are trained from scratch. All other LLM parameters remain frozen.

**Tokenizer.** We experimented with adding AMR role labels (e.g., `:ARG0`, `:ARG1`, `:mod`) to the tokenizer and extending the model's vocabulary accordingly. However, we found that the default tokenizer yielded more stable performance, suggesting that extending the vocabulary with role labels did not offer additional benefits. Therefore, we retain the original tokenizer throughout all experiments.

**Hardware setup.** All models were trained on a single GPU node with 64 GB of RAM. Models with 2 billion parameters or more were trained on an NVIDIA H100 GPU, while smaller models ($< 2$B parameters) were trained on an A100 GPU.

### B.3.1 Hyperparameters

The hyperparameter choice for each model can be found in Table 4 and Table 5.

**LoRA Hyperparameters.** Given the high computational cost of fine-tuning, we adopted a practical manual hyperparameter search strategy focused on LoRA configurations. We used all LoRA layer types (query, key, value, projection, and head) by default, with a rank ($r$) and scaling factor ($\alpha$) chosen from {4, 8, 16, 32}. Dropout rates were selected from {0.05, 0.1, 0.15}. In cases where overfitting was observed, we first

---

[4] https://github.com/Lightning-AI/litgpt

Table 4: Hyperparameter configurations for each SAFT models

| Category | Hyperparameter | LLaMA 1B | LLaMA 3B | Qwen 0.5B | Qwen 1.5B | Qwen 3B | Gemma 2B |
|---|---|---|---|---|---|---|---|
| LoRA | Rank ($r$) | 32 | 32 | 32 | 32 | 16 | 32 |
| | Scaling factor ($\alpha$) | 32 | 64 | 32 | 64 | 16 | 32 |
| | Dropout | 0.05 | 0.05 | 0.05 | 0.05 | 0.05 | 0.05 |
| | Head enabled | True | True | True | True | True | True |
| Training | Epochs | 6 | 5 | 6 | 5 | 6 | 5 |
| | Warmup steps | 100 | 100 | 100 | 100 | 100 | 100 |
| | Effective batch size | 256 | 256 | 256 | 256 | 256 | 256 |
| Custom | # Eigenvectors ($k$) | 30 | 30 | 30 | 30 | 30 | 25 |
| | MLP LR Multiplier ($\mu$) | 0.8 | 0.8 | 0.9 | 0.8 | 0.8 | 0.5 |
| | Magnetic param ($q$) | 0.25 | 0.25 | 0.25 | 0.25 | 0.25 | 0.25 |
| | Sinusoidal base ($q_{\sin}$) | 1000 | 1000 | 1000 | 1000 | 1000 | 1000 |
| | Sinusoidal dim ($d$) | 8 | 8 | 8 | 8 | 8 | 8 |

Table 5: Hyperparameter configurations for each conventionally fine-tuned model.

| Category | Hyperparameter | LLaMA 1B | LLaMA 3B | Qwen 0.5B | Qwen 1.5B | Qwen 3B | Gemma 2B |
|---|---|---|---|---|---|---|---|
| LoRA | Rank ($r$) | 16 | 8 | 16 | 32 | 16 | 32 |
| | Scaling factor ($\alpha$) | 16 | 8 | 16 | 32 | 16 | 32 |
| | Dropout | 0.05 | 0.05 | 0.05 | 0.05 | 0.05 | 0.05 |
| | Head enabled | True | True | True | True | True | True |
| Training | Epochs | 5 | 5 | 8 | 6 | 8 | 5 |
| | Warmup steps | 100 | 100 | 100 | 100 | 100 | 100 |
| | Effective batch size | 256 | 256 | 256 | 256 | 256 | 256 |

adjusted the dropout rate to improve generalization. If overfitting persisted, we disabled the LoRA head component, which we found to be the least critical for performance in preliminary runs. This strategy allowed us to balance empirical effectiveness with computational feasibility. Ranks were tuned independently per condition to maximize each method's performance.

**Epochs and training time.** All models were trained for 10 epochs with checkpoints saved at the end of each epoch, and the one with the best validation BLEU was chosen (the number of epochs reported is the one with the best BLEU). Training time varied from 9 hours for the smallest models to 16 hours for the larger ones.

**Leaning rate.** We use a learning rate schedule with linear warmup for the first 100 optimizer steps, followed by cosine annealing until the end of training.

**Custom hyperparameters.** There are five hyperparameters that are specific to our approach:

- **Number of eigenvectors** ($k$): the number of eigenvectors used as positional encodings; we select the $k$ eigenvectors corresponding to the smallest $k$ eigenvalues. We found that the performance is most stable in the range of 20 to 40 eigenvectors and therefore we chose from {20,25,30,35,40}.

- **MLP learning rate multiplier** ($\mu$): to improve training stability, we scale the learning rate of the MLP projecting positional encodings by a constant factor $\mu$, applied on top of the scheduled learning rate; that is, $\mathrm{LR}_{f_\theta}(t) = \mu \cdot \mathrm{LR}(t)$, where $\mathrm{LR}(t)$ is the base learning rate at step $t$.

- **Magnetic parameter** ($q$): controls the strength of the complex rotation in the magnetic Laplacian, modulating the influence of edge directionality. After experimenting with values between $10^{-3}$ and 0.5, we found $q = 0.25$ yielded the most stable results and fixed it for most experiments.

- **Sinusoidal PE frequency base** ($q_{\sin}$): the base used in the frequency scaling of sinusoidal positional encodings, analogous to that in Transformer models. Since inter-node sequences are relatively short in our setting, we use $q_{\sin} = 1000$.

- **Sinusoidal PE dimension** ($d$): defines the number of features in the sinusoidal positional encodings concatenated with the eigenvector-based encodings. We set this to 8.

**Models.** We used Low-Rank Adaptation (LoRA) (Hu et al., 2022) to fine-tune the following pretrained LLMs: LLaMA 3.2 (3B and 1B) (Touvron et al., 2023), Qwen 2.5 (3B, 1.5B, and 0.5B) (Bai et al., 2023), Gemma 2B (Team et al., 2024). Fine-tuning Gemma 7B was infeasible on the full dataset: a significant fraction of document-level AMR sequences exceed Gemma 7B's maximum context window, causing activation memory to grow unboundedly during the forward pass and resulting in out-of-memory errors that could not be resolved by reducing batch size alone. For each model, we compare two variants: one fine-tuned with our positional encodings (PEs) integrated during training, and one without. For both variants, we report results using the best-performing checkpoint found during development. During evaluation, the PEs are activated consistently based on the corresponding training configuration.

### B.3.2 Prompting Format

**Fine-tuning prompt.** To enable AMR-to-text generation with large language models, we adopt a structured prompting format implemented via the `AMR2Text` prompt style. Each prompt consists of three components:

- A **starting token**, which includes task metadata and generation instructions:

  ```
  <AMR-to-Text>
  [Task: AMR-to-Text]
  [Instruction] Convert the following AMR into natural language text.
  [Input: AMR]
  ```

- The **linearized AMR graph** $\mathcal{L}_{\mathcal{A}}$, inserted directly after the input header. This is a token sequence derived from the input AMR graph $\mathcal{A}$ (see Section 3.1).

- An **ending token**, marking the beginning of the generation segment:

  ```
  [Output: Text]
  ```

The full prompt passed to the model is thus structured as:

```
<AMR-to-Text>
[Task: AMR-to-Text]
[Instruction] Convert the following AMR into natural language text.
[Input: AMR]
𝓛𝒜
[Output: Text]
```

**Few-shot prompt for GPT models.** The full prompt passed to the model is structured as:

```
You are an assistant that converts AMRs to fluent English sentences.
Your job: For each item, convert its 'amr' into one fluent English sentence.
Do not include explanations; only output JSON.

Output format (STRICT):
{
"predictions": [ { "id": "...", "text": "..." } ]
}

### FEW-SHOT EXAMPLES
Example 1 (input): ["id":"1","amr":"(u/understand-01 :ARG0 (i/i)
```

```
:ARG1 (t/thing :ARG1-of (s/say-01 :ARG0 (p/person :name
(n/name :op1 "Ron" :op2 "Paul"))))"]
Example 1 (output): {"predictions":[{"id":"1","text":"I get what Ron Paul is
saying."}]}
Example 2 (input): ["id":"2","amr":"(s/say-01 :ARG0 (l/libertarian
:mod (h/hardcore)) :ARG1 (t/that))"]
Example 2 (output): {"predictions":[{"id":"2","text":"Thats what a Hardcore
Libertarian would say."}]}
Example 3 (input): ["id":"3","amr":"(h/hard-02 :degree (k/kind-of) :ARG1
(i/import-01 :ARG1 (f/food)) :ARG1-of (c/cause-01 :ARG0 (c2/consider-01
:ARG1 (p/probable :domain (w/wipe-out-02 :ARG1 (s/store) :mod (t/too))))))"]
Example 3 (output): {"predictions":[{"id":"3","text":"Kind of hard to import food
considering that the stores are probably wiped out too."}]}

...

### NOW PROCESS THE REAL BATCH
[ {"id":"0","amr":"(d/date-entity :day 21 :month 8 :year 2007)"} ]
```

## C  Additional Experiments

### C.1  Stratified evaluation over number of nodes

Similarly to the evaluation in Section 4.3, we perform a stratified analysis based on the number of nodes in the original AMR graph $\mathcal{A}$ to examine how both graph size and structural complexity influence the performance of SAFT. As shown in Figure 7, SAFT exhibits consistent improvements over standard fine-tuning across most model sizes, particularly for larger models. However, the trend is less pronounced than in Figure 2, where stratification was based on graph depth. This contrast highlights that the gains from SAFT are more strongly associated with structural complexity and long-range dependencies than with graph size alone, suggesting that structural information yields diminishing returns when applied to merely larger—but not necessarily deeper—graphs.

### C.2  Effect of model scale on SAFT

A common hypothesis is that structural inductive biases become less relevant as model scale increases, under the assumption that sufficiently large language models can internalize structural reasoning through parametric capacity alone. Our results do not provide evidence in support of this hypothesis within the range of model sizes we evaluate.

As shown in Figure 8, the relative improvements of SAFT over standard fine-tuning (FT) do not exhibit a monotonic decreasing trend with increasing model size. Instead, gains fluctuate across both scale and model family. For example, the largest BLEU improvements are observed for mid-sized LLaMA models, while the strongest ChrF improvements occur for the 3B Qwen model. Moreover, these trends are not consistent across metrics: within a single model family, BLEU and ChrF gains follow different trajectories as scale increases, and model rankings vary depending on the evaluation measure.

We emphasize that this does not rule out the possibility that, at larger scales than those considered in this study, the benefits of SAFT may eventually diminish or disappear. However, within the regimes we test, no such pattern is observable. Instead, the effect of SAFT appears to depend on a combination of model scale, architecture, and generation dynamics, rather than scale alone.

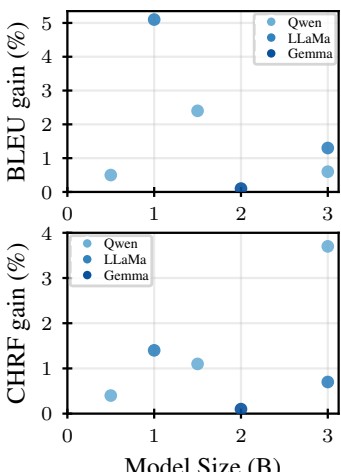

Figure 8: Relative BLEU and ChrF++ improvements as a function of model size.

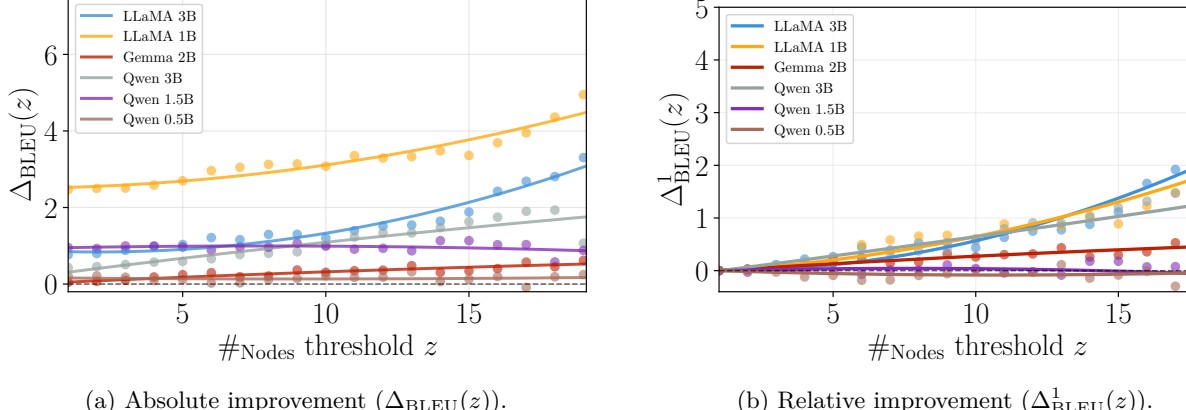

(a) Absolute improvement ($\Delta_{\text{BLEU}}(z)$).       (b) Relative improvement ($\Delta^1_{\text{BLEU}}(z)$).

Figure 7: BLEU score improvements of structurally-aware fine-tuned (SAFT) models over conventionally fine-tuned (FT) counterparts, on AMR instances with number of nodes $\#_{\text{Nodes}}(\mathcal{A}) \geq z$. (a) **Absolute improvement** ($\Delta_{\text{BLEU}}$): differences in BLEU score between SAFT and FT models across varying number of nodes and model families. (b) **Relative improvement** ($\Delta^1_{\text{BLEU}}$): differences in BLEU scores normalized by single-node graph performance. The results show consistent gains, though the magnitude of improvement is less pronounced compared to depth-based stratification (see Figure 2), indicating that structural complexity plays a more critical role than graph size alone. All lines are second-degree polynomial fits.

Therefore, based on current evidence, the claim that increasing model size renders SAFT unnecessary is not supported.

### C.3 Bootstrap paired significance test

To evaluate whether the performance differences between SAFT and standard fine-tuning (FT) are robust to test-set variation, we conduct a bootstrap paired significance analysis. For each model, we repeatedly resample the AMR 3.0 test set with replacement and compute BLEU, and ChrF for both FT and SAFT on each bootstrap replicate. This yields paired score distributions without requiring multiple independent training runs, allowing us to isolate test-set variability as a source of uncertainty.

Table 6 reports the bootstrap mean and standard deviation for each metric and model. Across all settings, the variance induced by resampling is small (typically within $\pm 0.3$–$0.5$ BLEU), indicating that the observed performance differences are not driven by unstable test-set fluctuations. Bootstrap means differ slightly from Table 1 due to resampling variability; point estimates remain the reference.

To further characterize the consistency of these differences, Figure 9 presents a win-rate matrix, showing for each model–metric pair the fraction of bootstrap samples in which SAFT outperforms FT. In most cases, SAFT wins on a clear majority of samples, including all LLaMA variants and two out of the three Qwen models. For settings where the average improvements are small, the bootstrap distributions reflect this appropriately through near-balanced win rates, rather than artificially inflating significance.

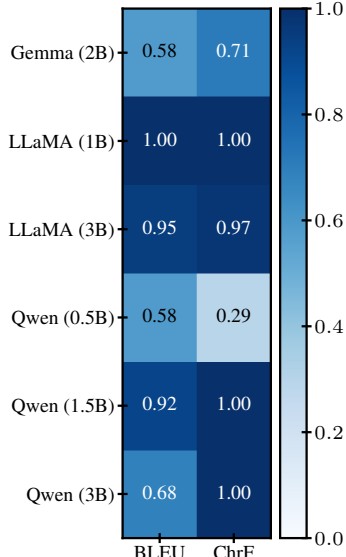

Figure 9: Win-rate heatmap of the bootstrap paired test.

Overall, this analysis indicates that the gains reported in the main results are stable under reasonable test-set perturbations. At the same time, the bootstrap results remain appropriately conservative in cases where the FT–SAFT gap is minimal, avoiding overinterpretation of marginal differences.

Table 6: **Bootstrap means and standard deviations of BLEU, and ChrF over 50 resampled test sets.** The variability is small across models, and the relative behavior of SAFT vs. FT aligns with the trends reported in the main results. Bold results are statistically significant ($p < 0.05$).

| Model | Variant | BLEU | ChrF |
|---|---|---|---|
| Gemma (2B) | FT | $52.97 \pm 0.49$ | $69.74 \pm 0.29$ |
| | SAFT | $53.01 \pm 0.48$ | $69.98 \pm 0.30$ |
| LLaMA 3.2 (1B) | FT | $45.64 \pm 0.55$ | $65.74 \pm 0.37$ |
| | SAFT | $\mathbf{47.75 \pm 0.54}$ | $\mathbf{67.65 \pm 0.33}$ |
| LLaMA 3.2 (3B) | FT | $53.52 \pm 0.40$ | $71.70 \pm 0.25$ |
| | SAFT | $54.45 \pm 0.44$ | $72.40 \pm 0.30$ |
| Qwen 2.5 (0.5B) | FT | $42.68 \pm 0.40$ | $63.14 \pm 0.30$ |
| | SAFT | $42.94 \pm 0.52$ | $62.94 \pm 0.30$ |
| Qwen 2.5 (1.5B) | FT | $50.66 \pm 0.46$ | $68.69 \pm 0.30$ |
| | SAFT | $51.69 \pm 0.54$ | $\mathbf{70.70 \pm 0.34}$ |
| Qwen 2.5 (3B) | FT | $51.75 \pm 0.50$ | $69.33 \pm 0.37$ |
| | SAFT | $52.00 \pm 0.45$ | $\mathbf{70.44 \pm 0.29}$ |

Table 7: **Comparison of FT vs. SAFT.** We report METEOR and BERTScore for Qwen 2.5 and LLaMA 3.2 (3B) on AMR 3.0. Best results per metric are highlighted in bold, with relative gain in parentheses.

| Model | Variant | METEOR ↑ | BERTScore ↑ |
|---|---|---|---|
| Qwen 2.5 (3B) | FT | 59.4 | 82.85 |
| | SAFT | **60.1** (+1.18%) | **83.69** (+1.01%) |
| LLaMA 3.2 (3B) | FT | 70.3 | 86.19 |
| | SAFT | **70.5** (+0.28%) | **86.22** (+0.03%) |

Table 8: **Impact of graph depth on gains.** We report relative improvements in BERTScore and METEOR across different depths.

| Graph Depth | BERTScore Gain ↑ | METEOR Gain ↑ |
|---|---|---|
| 0 | 0.84 | 1.08 |
| 3 | 1.30 | 1.31 |
| 9 | 3.11 | 5.01 |
| 10 | 2.96 | 3.27 |

## C.4 Extra Evaluation Metrics

Along with the metrics reported in Section 4.2, we evaluate SAFT vs. FT using METEOR (Lavie & Agarwal, 2007) and BERTScore (Zhang et al., 2020), two complementary metrics that are more sensitive to semantic adequacy and fluency. The results are shown in Table 7. We further stratified the gains by AMR graph depth, following the same setup as in Figure 2. Both BERTScore and METEOR gains increase with graph depth as shown in Table 8. These results support our claim: SAFT becomes increasingly beneficial for structurally complex inputs, and these improvements hold across multiple metrics.

## C.5 Runtime

We provide a formal breakdown of SAFT's time and space complexity and clarify the scope of its computational overhead. All graph-related operations are performed once at preprocessing time and do not affect training or inference time. These include: graph preprocessing, magnetic Laplacian computation, partial eigendecomposition (EVD), and positional encoding computation. The main bottleneck in this process is the partial EVD.

The dense partial EVD requires $\mathcal{O}(kn^2)$, where $k$ is the number of computed eigenvalues, $n$ the number of nodes. For most practical scenarios in the AMR-to-text generation task, this complexity is manageable, as AMRs are relatively small, with AMR 3.0 ($\sim$54 nodes) and DocAMR ($\sim$730 nodes). Asymptotically, sparse solvers become advantageous when $n \gg k$. This translates in practice to $n \gtrsim 2000$. In this case, the complexity is $\mathcal{O}(km)$ where $m$ is the number of edges.

Table 9: **Sparse vs. dense preprocessing times.** We report average and maximum runtimes (in seconds) for AMR 3.0 and DocAMR.

| Dataset | Method | Avg Time (s) ↓ | Max Time (s) ↓ |
|---------|--------|----------------|----------------|
| AMR 3.0 | Sparse | 0.009 | 0.12 |
|         | Dense  | 0.027 | 0.57 |
| DocAMR  | Sparse | 1.31  | 12.51 |
|         | Dense  | 0.28  | 2.49 |

Table 10: **Inference and preprocessing times.** Inference time and graph preprocessing time (including EVD, sinusoidal encoding, and projection via the MLP) for Qwen 3B.

| Dataset | Inference Time (s) | | Preprocessing Time (s) | |
|---------|---------|---------|---------|---------|
| AMR 3.0 | Avg: 1.73 | Max: 6.81 | Avg: 0.01 | Max: 0.12 |
| DocAMR | Avg: 325.24 | Max: 582.22 | Avg: 0.28 | Max: 2.49 |

We first show in Table 9 that, in our case, using dense solvers yields faster runtime compared to sparse solvers. Then, we report in Table 10 the average and maximum time (in seconds) required for inference and precomputation, showing the negligible impact of the latter. The computation of our structure-aware encodings is a preprocessing step that leaves the model architecture and runtime efficiency intact. Since structure-aware encodings are computed once per graph and reused throughout training and inference, the overall overhead remains minimal.

At inference time, SAFT does not change the computation inside the Transformer itself. The token sequence length and the hidden dimensionality remain the same, so the attention and feed-forward layers have the same cost as in standard fine-tuning (FT). The only additional costs compared to FT come from two sources. First, we compute the graph positional encodings for the input AMR. This is done once per graph and does not depend on the decoding length. As shown in Table 10, this step adds on average 0.01 seconds on AMR 3.0 and 0.28 seconds on DocAMR, which is small compared to the overall inference time. Second, we apply a lightweight two-layer MLP to project these encodings into the model's embedding space. This MLP maps $\mathbb{R}^{2k+d} \to \mathbb{R}^{d_{\text{emb}}}$ and $\mathbb{R}^{d_{\text{emb}}} \to \mathbb{R}^{d_{\text{emb}}}$, and introduces $d_{\text{emb}}(2k + d + d_{\text{emb}} + 2)$ additional parameters. This overhead is negligible relative to the size of the underlying LLM.

### C.6 Potential applicability beyond AMR

SAFT does not assume properties unique to AMR and only requires a graph structure together with a node-to-token alignment. This suggests a potential extensions to other graph-structured data associated with text, such as semantic role graphs, knowledge graph triplets, or discourse trees mapped to sentences, just to name a few. A full empirical study of these settings is outside the scope of this work, but we view this as a promising direction for future research.

## D Limitations

While SAFT achieves consistent improvements, particularly on semantically complex inputs, several limitations remain.

First, all experiments are conducted using LoRA-based fine-tuning on models up to 3B parameters. Whether SAFT's gains persist under full fine-tuning, or at larger scales (e.g., 7B or beyond), remains an open empirical question. Within the 0.5B–3B range we evaluate, gains do not exhibit a monotonic relationship with scale (Appendix C.2), but we cannot extrapolate beyond this regime. Second, all empirical evaluation is limited to AMR-to-text generation. While SAFT is conceptually applicable to any task with directed graph inputs and node–token alignment, generalization to other tasks, such as semantic role graphs, knowledge graph verbalization, or discourse-to-text, has not been empirically verified. Third, gains are less pronounced on simpler inputs with limited structural complexity, suggesting the method's inductive bias is not universally beneficial. Fourth, effectiveness depends on hyperparameter choices such as positional encoding dimensionality $k$ and the magnetic phase parameter $q$, which may require task-specific tuning. Finally, extending SAFT to new tasks requires defining a node-to-token alignment, which adds modest engineering effort.

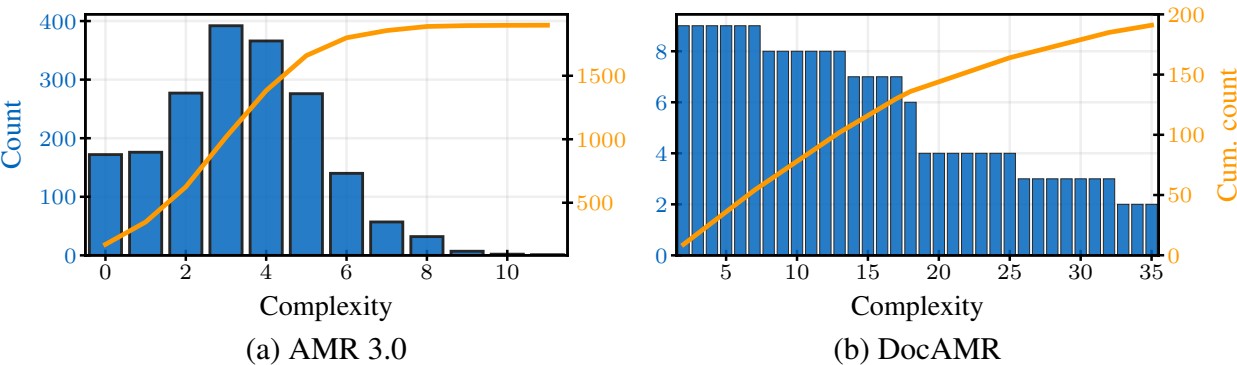

Figure 10: Count of AMR graphs per complexity on both AMR 3.0 and DocAMR (test sets).

## E  Assets and Licences

### E.1  Datasets

We evaluate our approach on the AMR 3.0 dataset (LDC2020T02[5]) (Knight et al., 2020), which consists of approximately 55k training instances, 1.3k for development, and 1.4k for testing. Compared to earlier versions (Knight et al., 2017), AMR 3.0 includes more diverse graph structures and broader linguistic coverage, providing a rigorous benchmark for AMR-to-text generation.

This release is a semantically annotated corpus of over 59k English sentences drawn from a diverse mix of domains, including broadcast conversation, discussion forums, weblogs, newswire, and fiction. Annotations cover PropBank-style frames, non-core semantic roles, coreference, named entities, modality, negation, quantities, and questions. Sentence-level annotations are represented as rooted, directed acyclic graphs designed to abstract away from surface syntax and emphasize predicate-argument structure.

For a subset of experiments, we also evaluate on the DocAMR dataset (part of AMR 3.0), which extends AMR to the document level by providing inter-sentence coreference and discourse-level annotations. This enables assessment of long-range semantic dependencies and coherence in multi-sentence generation. DocAMR consists of 284 documents in the training split and 9 documents in the test split, covering a total of 8027 gold sentence-level AMRs.

We use the dataset as released by the Linguistic Data Consortium (LDC2020T02), without augmenting with any silver data (i.e., data labeled through heuristic or automated methods). AMR 3.0 is distributed under the LDC User Agreement and is not publicly available; access requires an institutional or individual LDC license. For reference, the release was published on January 15, 2020 and includes contributions from DARPA-funded programs (BOLT, DEFT, MRP, LORELEI) and NSF-supported research.

To make sample availability across structural complexities explicit, we include histograms of graph depths for AMR 3.0 and DocAMR (Figure 10). AMR 3.0 exhibits a long tail with few very high-depth graphs, whereas DocAMR contains nine document-level graphs whose constituent sentence-level AMRs all originate from the AMR 3.0 test split, each covering many AMRs. To test the capabilities of our model across multiple complexities (number of AMRs in the case of DocAMR), we sample subgraphs from document-level AMRs of increasing complexities, with a resulting count distribution as shown in Figure 10(b).

Table 11: Datasets used for AMR-to-text generation.

| Dataset | Size (Train/Dev/Test) | Key Features | License |
|---|---|---|---|
| AMR 3.0 | 55k / 1.3k / 1.4k | Sentence-level graphs, broad linguistic coverage | LDC Non-Member License |
| DocAMR | 284 / - / 9 | Document-level annotations, coreference, discourse | LDC Non-Member License |

---

[5] https://catalog.ldc.upenn.edu/LDC2020T02

### E.2 Models

We conduct experiments using a selection of publicly available pretrained language models with open or research-focused licenses. All models are used strictly for academic purposes, in compliance with their respective licenses.

**LitGPT.** We build on the `LitGPT` framework[6], an open-source project released under the Apache License 2.0[7]. It provides modular components for efficient fine-tuning, inference, and reproducibility across large-scale models.

**Qwen.** Qwen models[8], developed by Alibaba Cloud, are released under the Apache License 2.0. This permissive open-source license permits modification, distribution, and commercial use, provided appropriate attribution is maintained.

**Gemma.** Gemma[9], developed by Google DeepMind, is also licensed under the Apache License 2.0. This allows for both academic and commercial applications and emphasizes interoperability with a wide range of open-source software.

**LLaMA 2.** LLaMA 2 models[10], released by Meta, are governed by the LLAMA 2 Community License Agreement. The license permits use, modification, and redistribution, but restricts:

- Commercial use by entities exceeding 700 million monthly active users without explicit permission from Meta

- Use of LLaMA outputs to train competing large language models

Redistributions must include a notice file, and use is subject to Meta's Acceptable Use Policy[11].

Table 12: Pretrained models and licensing details.

| Model | Provider | License | Notes |
|---|---|---|---|
| LitGPT | Lightning AI | Apache 2.0 | Permissive, for training and inference |
| Qwen | Alibaba Cloud | Apache 2.0 | Open-source, commercial use permitted |
| Gemma | Google DeepMind | Apache 2.0 | Open-source, commercial use permitted |
| LLaMA 2 | Meta | LLAMA 2 Community License | Requires license for large-scale commercial use |

## F Additional information on related work

Prior work on AMR-to-text generation—and more broadly, text generation from graph-structured data—has been explored through three main paradigms: **linearization-based approaches**, which serialize graphs into sequences; **adapter-based approaches**, which introduce graph-native modules into pretrained architectures; and **graph-tuning-based approaches**, which couple LLMs with trainable graph encoders or embeddings. To the best of our knowledge, no prior work has investigated graph positional encodings as a lightweight, architecture-agnostic means of enabling structure-aware fine-tuning.

---

[6]https://github.com/Lightning-AI/litgpt
[7]http://www.apache.org/licenses/LICENSE-2.0
[8]https://github.com/QwenLM/Qwen
[9]https://github.com/google-deepmind/gemma
[10]https://ai.meta.com/resources/models-and-libraries/llama-downloads/
[11]https://llama.com/use-policy

### F.1 Linearization-based approaches

These methods convert the input graph into a linear sequence and fine-tune a pre-trained encoder-decoder transformer (e.g., BART, T5) in a standard seq-to-seq setup.

Bevilacqua et al. (2021) introduced a symmetric framework for AMR parsing and generation by fine-tuning BART on linearized AMR graphs using both DFS and BFS traversals (SPRING). AMR-BART (Bai et al., 2022) builds on SPRING by incorporating self-supervised graph denoising tasks during pretraining, which improves robustness to structural noise. BiBL (Cheng et al., 2022) further extends this line of work by jointly modeling AMR-to-text and text-to-AMR transitions through single-stage multitask learning with auxiliary losses. These models share a common foundation: they linearize the AMR graph and fine-tune a standard transformer. This strategy has also been applied to large language models (LLMs) via fine-tuning (Raut et al., 2025; Mager et al., 2020) or prompting (Yao et al., 2024a; Jin et al., 2024b) using the linearized AMR graph as input.

More generally, the practice of aligning LLMs with structured data through linearization has found success across domains such as molecular generation (Zheng et al., 2024), network traffic analysis (Cui et al., 2025), tabular reasoning (Fang et al., 2024), and 3D mesh processing (Wang et al., 2024).

### F.2 Adapter-based approaches

Adapter-based methods directly model the structure of the input graph using graph neural networks (GNNs) or related components, which are then integrated into transformer architectures.

StructAdapt (Ribeiro et al., 2021) is built around the encoder–decoder architecture (e.g., T5) in which a dedicated graph encoder (which they call StructAdapt) produces structure-enriched representations that the decoder then consumes via cross-attention and MLP adapters. Its core design relies on this asymmetry: graph processing happens entirely in the encoder, while the decoder only receives those enriched encoder states. Other methods take a similar direction by modifying the attention mechanism to incorporate structural biases from the input graph (Zhu et al., 2019). Another line of work avoids transformer pretraining altogether, instead training graph-to-sequence models from scratch that can natively process graph inputs (Song et al., 2018; Wang et al., 2020).

### F.3 LLMs for graph-structured data

The rise of large language models (LLMs) (Vaswani et al., 2017; Devlin et al., 2019; Brown et al., 2020; Touvron et al., 2023) has reshaped NLP. Recently, there has been growing interest in extending LLMs to handle graph-structured inputs (Jin et al., 2024a), particularly in domains like molecules, knowledge graphs, and social networks. Existing methods typically fall into one of three strategies: (i) flattening graphs into linear sequences (Jiang et al., 2023; Fatemi et al., 2024; Yao et al., 2024b); (ii) modifying the LLM architecture to incorporate graph encoders (Zhang et al., 2022); or (iii) generating structure-aware token embeddings that align with LLM representations (Tian et al., 2024; Tang et al., 2024). The latter direction shows promise but introduces additional training complexity due to the need for separate graph encoders and alignment mechanisms.

### F.4 Graph-tuning-based approaches

A more recent line of work integrates graph modules directly into LLMs, aiming to couple structural encoding with pretrained language models. GraphAdapter (Huang et al., 2024) employs a trainable GNN as an adapter, aligned with the LLM during fine-tuning, requiring both pretraining and parameter-intensive alignment. GraphLLM (Chai et al., 2023) inserts graph transformers at every layer of the LLM by introducing learned graph-based prefix tokens into the key/value projections. GraphGPT (Tang et al., 2024) prepends graph embeddings produced by a trainable GNN into the prompt via a learned projector, keeping the LLM frozen. While these methods inject structural information effectively, they introduce substantial complexity in the form of additional trainable modules, architectural modifications, or expensive pretraining requirements.

## F.5 Comparison and positioning of SAFT

Across these paradigms, a common trade-off emerges: linearization-based methods retain compatibility with standard architectures but discard explicit structure; adapter-based methods preserve structure but sacrifice full LLM compatibility; graph-tuning-based methods tightly integrate graphs with LLMs but at the cost of additional parameters and architectural changes. In contrast, SAFT is architecture-agnostic and parameter-efficient. It introduces relational inductive bias through precomputed, parameter-free positional encodings derived from the magnetic Laplacian spectrum. These encodings are aligned with the LLM's embedding space via a lightweight projection layer, requiring no graph-specific training, no architectural changes, and no costly pretraining. This design makes SAFT a simple and efficient way to enable structure-aware fine-tuning of LLMs, while remaining general to tasks involving graph-structured inputs.

# G Generated outputs and error analysis

We present a qualitative error analysis comparing baseline fine-tuning (FT) and our structure-aware fine-tuning method (SAFT). To avoid cherry-picking and to ensure that examples are sampled in a principled way, we compute smoothed sentence-level BLEU scores for both models on every test instance. We then partition the dataset into performance buckets based on percentile thresholds: cases where SAFT is considerably better than FT, cases where FT is considerably better than SAFT, and cases where both models either perform well or fail. From each bucket, we draw a small random sample of five examples. For each selected instance, we report the reference output, the two model predictions, and their respective BLEU scores. We additionally highlight token-level differences using a color-coded character diff to make divergences visually salient. The goal is not to claim statistical significance from individual examples, but to give the reader concrete insight into the characteristic strengths and failure modes of each system across distinct performance regimes.

**SAFT better**

**Ground Truth:** International; weapons; proliferation; Government; energy
**FT Prediction:** International; weapons; proliferation; Governmentergy; energy [BLEU: 28.6]
**SAFT Prediction:** International; weapons; proliferation; Government; energy [BLEU: 100.0]

---

**Ground Truth:** International; crime; Government; narcotics
**FT Prediction:** International; crime; gGovernment; narcotics [BLEU: 18.8]
**SAFT Prediction:** International; crime; Government; narcotics [BLEU: 100.0]

---

**Ground Truth:** The issues have been unresolved for 4 years.
**FT Prediction:** The issues werhave beenot unresolved for four4 years. [BLEU: 7.0]
**SAFT Prediction:** The issues have been unresolved for 4 years. [BLEU: 70.7]

---

**Ground Truth:** You can't get her sectioned for that.
**FT Prediction:** You can't get aher sectioned forbecausebecause that. [BLEU: 14.8]
**SAFT Prediction:** You can't get her sectioned for that. [BLEU: 100.0]

---

**Ground Truth:** 2. Create a few nuclear-powered aircraft carrier battle groups.
**FT Prediction:** 2. Createing a few nuclear-powered aircraft carriers groupsbattle groups. [BLEU: 5.6]
**SAFT Prediction:** 2. Create a few nucbattlear-powered aircraft carrier battle groups. [BLEU: 59.7]

---

**FT better**

**Ground Truth:** Ukraine does not supply or have plans to supply any armaments to the Government of South Sudan.
**FT Prediction:** Ukraine hadoes not supply orany have plans to supply any armsaments to the SGouvernmenth of South Sudan. [BLEU: 34.7]
**SAFT Prediction:** Ukraine hadoes not supply orany have planys to supply any armsaments to the SGouvernmenth of South Sudan. [BLEU: 12.6]

---

**Ground Truth:** The proposal may complicate the Bush administration's efforts to win an exemption for India to engage in nuclear trade.
**FT Prediction:** The proposedal maycould complicate the Bush administration's efforts to win Iandia exemption for India to engage in nuclear trade. [BLEU: 68.6]
**SAFT Prediction:** The proposedal maycould complicate the Buadminishtration administration's efforts to win Indian exemption forom India tfrom engage in nuclear trade. [BLEU: 29.7]

---

**Ground Truth:** In Virginia , it 's to benefit private business plans and not to serve the public interest .
**FT Prediction:** In Virginia ,it it 'hass ftor benefit privathe business plans ,and not to serve the public 'interest . [BLEU: 28.3]
**SAFT Prediction:** Itn Virginia ,it it 'iss ftor benefit privathe business plans ,and not to serve thepublic public 'interest . [BLEU: 8.4]

---

**Ground Truth:** 26/02/2010 14:32
**FT Prediction:** 26/02/2010 14:32 [BLEU: 31.6]
**SAFT Prediction:** 26February/02/2010 14:32 [BLEU: 15.0]

---

**Ground Truth:** Nepal (NP)
**FT Prediction:** Nepal (NP) [BLEU: 31.6]
**SAFT Prediction:** Nepal (NEP) [BLEU: 15.0]

---

**Both good**

**Ground Truth:** proliferation; technology; international; politics
**FT Prediction:** proliferation; technology; international; politics [BLEU: 100.0]
**SAFT Prediction:** proliferation; technology; international; politics [BLEU: 100.0]

---

**Ground Truth:** They deserve it. They asked for that.
**FT Prediction:** They deserve it. They asked for that. [BLEU: 100.0]
**SAFT Prediction:** They deserve it. They asked for that. [BLEU: 100.0]

---

**Ground Truth:** Tell your ex that all communication needs to go through the lawyer.
**FT Prediction:** Tell your ex that all communication needs to go through athe lawyer. [BLEU: 82.7]
**SAFT Prediction:** Tell your ex that all communication needs to go through athe lawyer. [BLEU: 82.7]

---

**Ground Truth:** Xinhua News Agency , Rome , September 1st , by reporters Aiguo Yang and Changrui Huang
**FT Prediction:** Xinhua News Agency , Rome , September 1st , by reporters Aiguo Yang and Changrui Huang [BLEU: 81.5]
**SAFT Prediction:** Xinhua News Agency , Rome , September 1st , by reporters Aiguo Yang and Changrui Huang [BLEU: 81.5]

---

**Ground Truth:** International; Government; technology; politics; economy
**FT Prediction:** International; Government; technology; politics; economy [BLEU: 100.0]
**SAFT Prediction:** International; Government; technology; politics; economy [BLEU: 100.0]

---

**Both bad**

**Ground Truth:** Haha
**FT Prediction:** Haha. [BLEU: 0.0]
**SAFT Prediction:** Haha, [BLEU: 0.0]

---

**Ground Truth:** Good Evening Digicel.
**FT Prediction:** Good eEvening,Digicel. [BLEU: 9.1]
**SAFT Prediction:** Good eEvening,Digicel,. [BLEU: 9.1]

---

**Ground Truth:** To help the survivors of the Gulf.
**FT Prediction:** ToHelp help those survivors inof the Gulf.region [BLEU: 3.7]
**SAFT Prediction:** ToSurv hsurvivelp those survivors inof the Gulf. [BLEU: 7.7]

---

**Ground Truth:** Please tell us our to pray because almost not believe in god and our prayer never arrived to god.
**FT Prediction:** Please tell us howur toprayers pray inwbecause wialmosthnout believeing in Ggod and heour prayers willnever arrived ato himgod. [BLEU: 6.2]
**SAFT Prediction:** Please,tell us howurto praybercauspray,we wealmostnot believeing in Ggod and our prayers willnever arrived to god. [BLEU: 3.4]

**Ground Truth:** You may think that 's not rational land use .
**FT Prediction:** You cmany think that i'snothrational ltheand use use. [BLEU: 5.4]
**SAFT Prediction:** You cmany think that 'snotratiosusingal land usinguse use. [BLEU: 5.7]

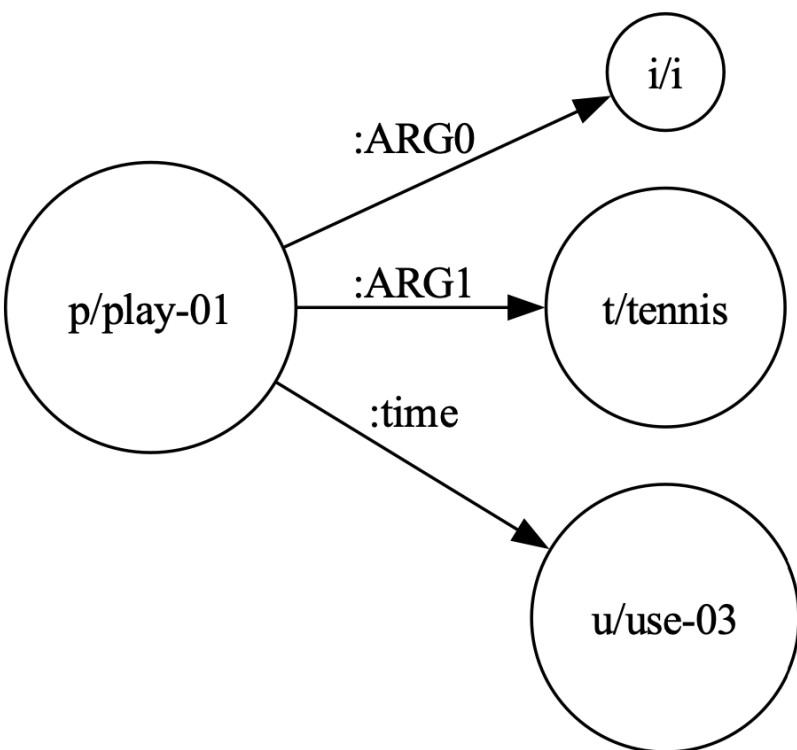

Figure 11: Original AMR graph for the sentence "I used to play tennis".

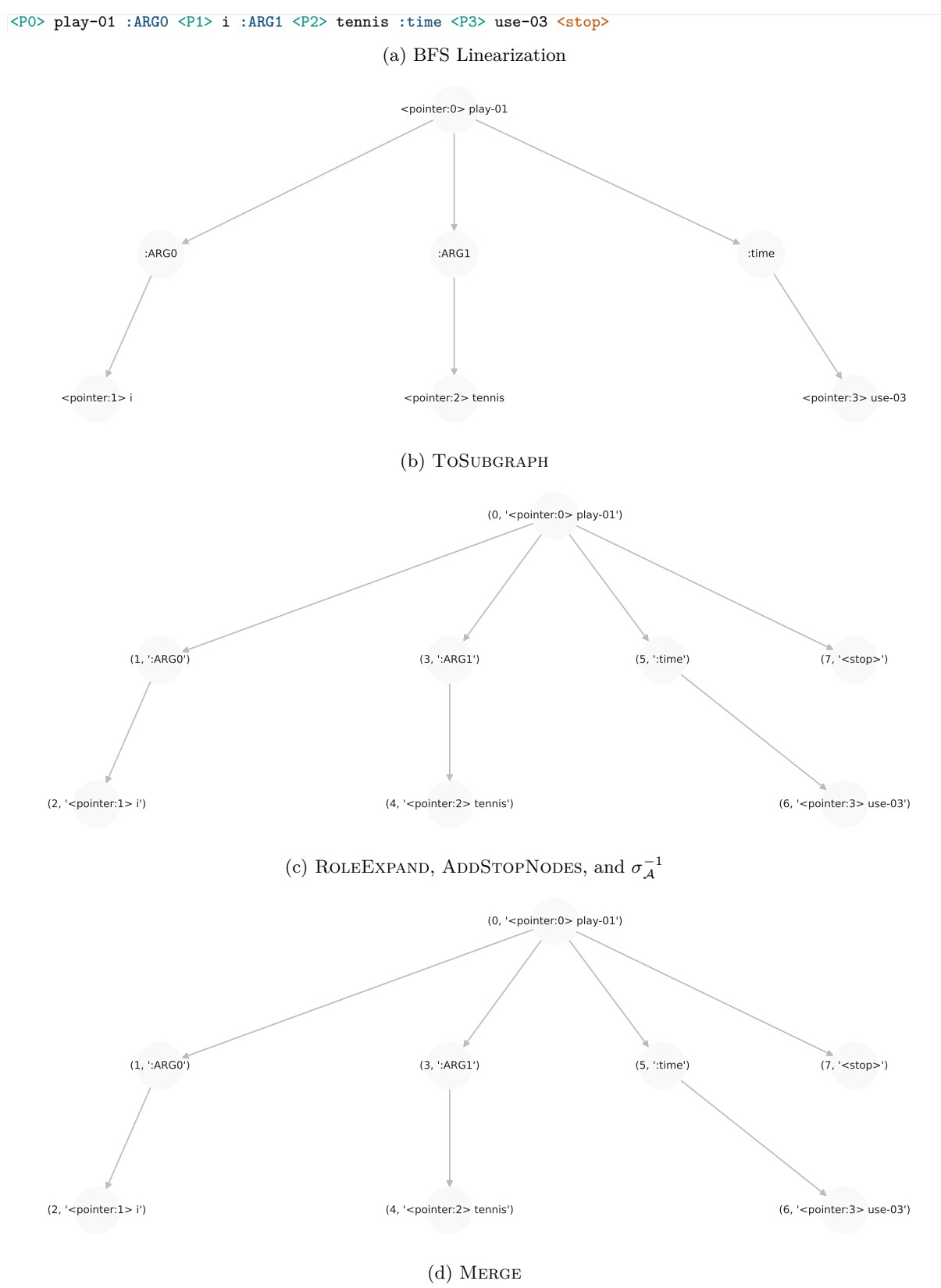

```
<P0> play-01 :ARG0 <P1> i :ARG1 <P2> tennis :time <P3> use-03 <stop>
```

(a) BFS Linearization

(b) ToSubgraph

(c) RoleExpand, AddStopNodes, and $\sigma_{\mathcal{A}}^{-1}$

(d) Merge

Figure 12: Overview of preprocessing steps for the AMR corresponding to the sentence: "I used to play tennis".

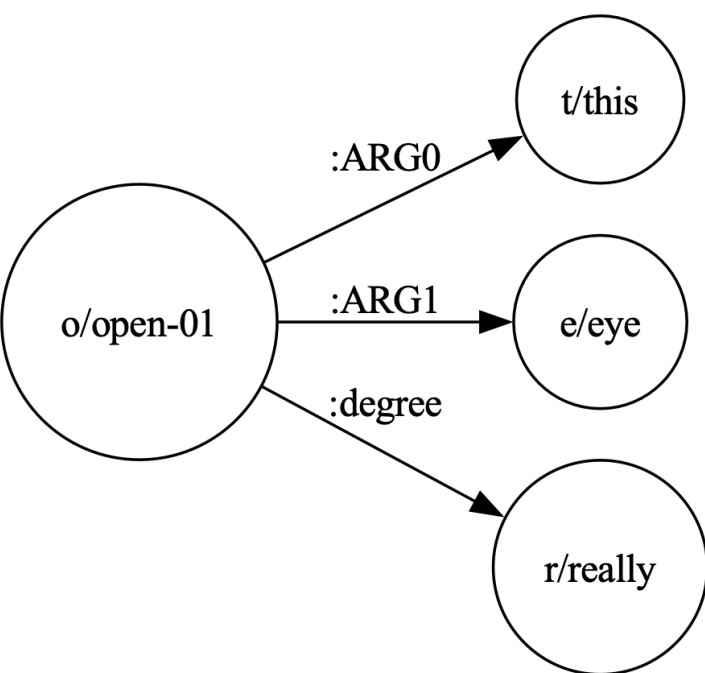

Figure 13: Original AMR graph for the sentence "This is really eye-opening".

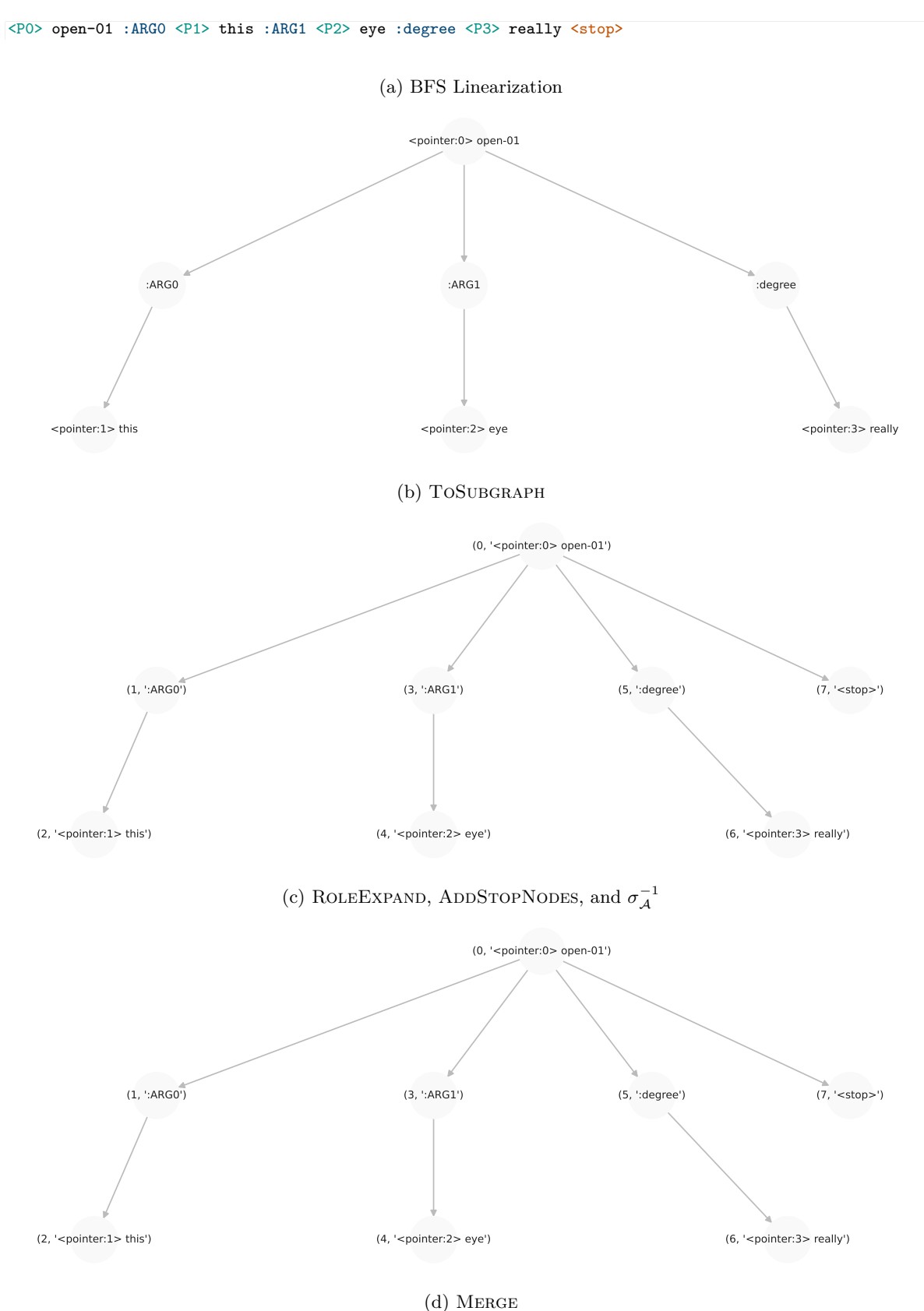

```
<P0> open-01 :ARG0 <P1> this :ARG1 <P2> eye :degree <P3> really <stop>
```

(a) BFS Linearization

(b) TOSUBGRAPH

(c) ROLEEXPAND, ADDSTOPNODES, and $\sigma_{\mathcal{A}}^{-1}$

(d) MERGE

Figure 14: Overview of preprocessing steps for the AMR corresponding to the sentence: "This is really eye-opening".

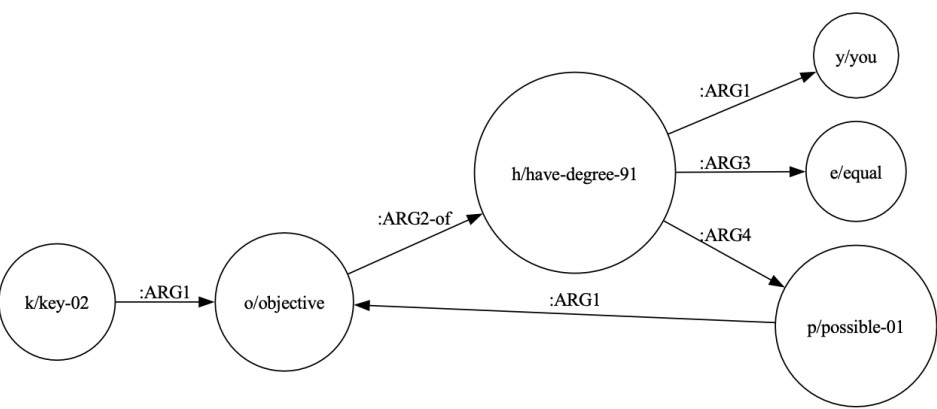

Figure 15: Original AMR graph for the sentence "The key is to be as objective as possible".

```
<P0> key-02 :ARG1 <P1> objective <stop> <P2> have-degree-91 :ARG2 <P1> :ARG1 <P3> you :ARG3 <P4> equal
:ARG4 <P5> possible-01 <stop> <P5> :ARG1 <P1> <stop>
```

(a) BFS Linearization

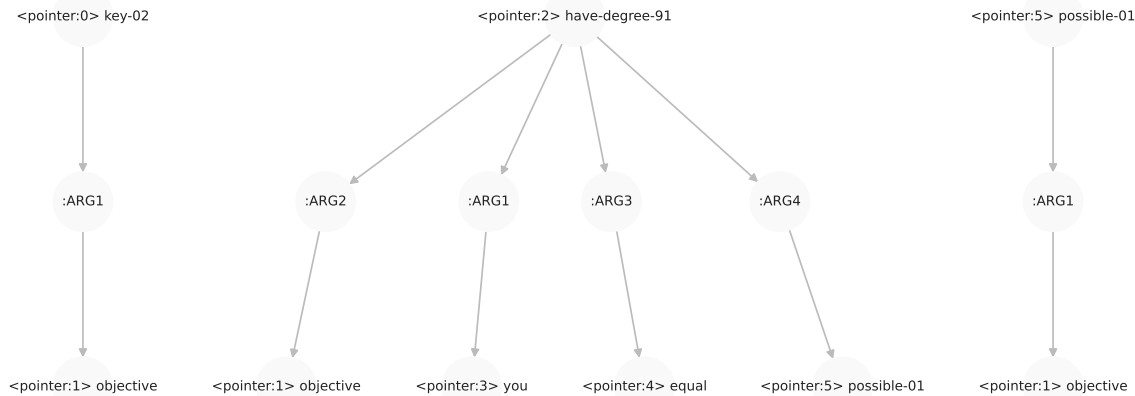

(b) TOSUBGRAPH

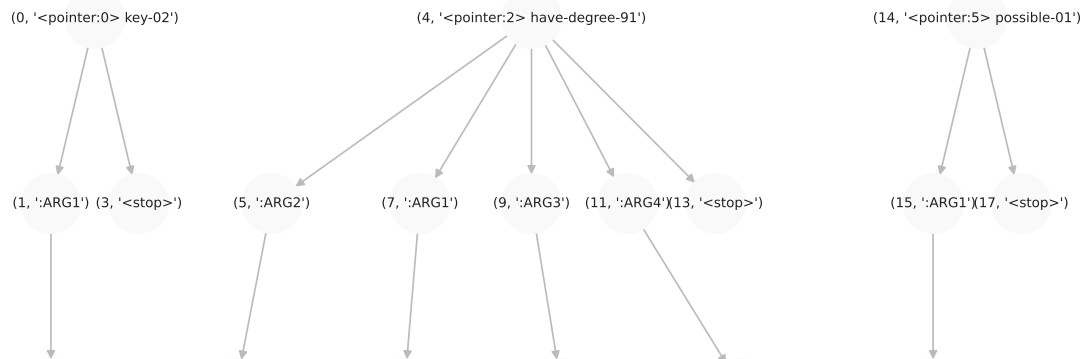

(c) ROLEEXPAND, ADDSTOPNODES, and $\sigma_{\mathcal{A}}^{-1}$

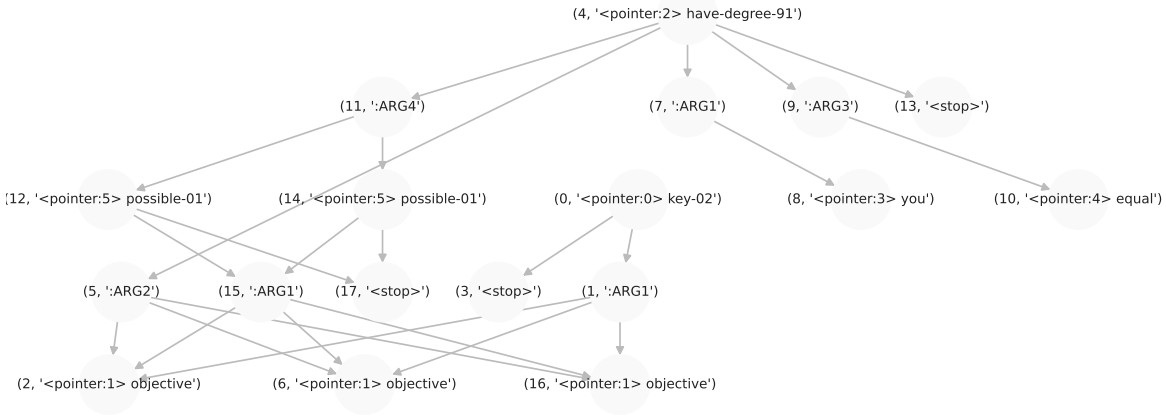

(d) MERGE

Figure 16: Overview of preprocessing steps for the AMR corresponding to the sentence: "The key is to be as objective as possible".

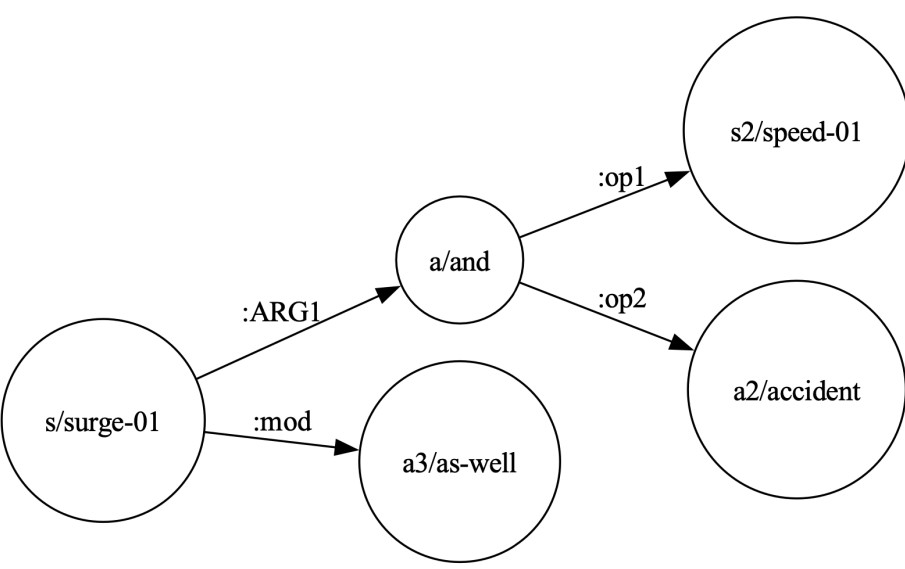

Figure 17: Original AMR graph for the sentence "Speeding and accidents have surged as well".

`<P0> surge-01 :ARG1 <P1> and :mod <P2> as-well <stop> <P1> :op1 <P3> speed-01 :op2 <P4> accident <stop>`

(a) BFS Linearization

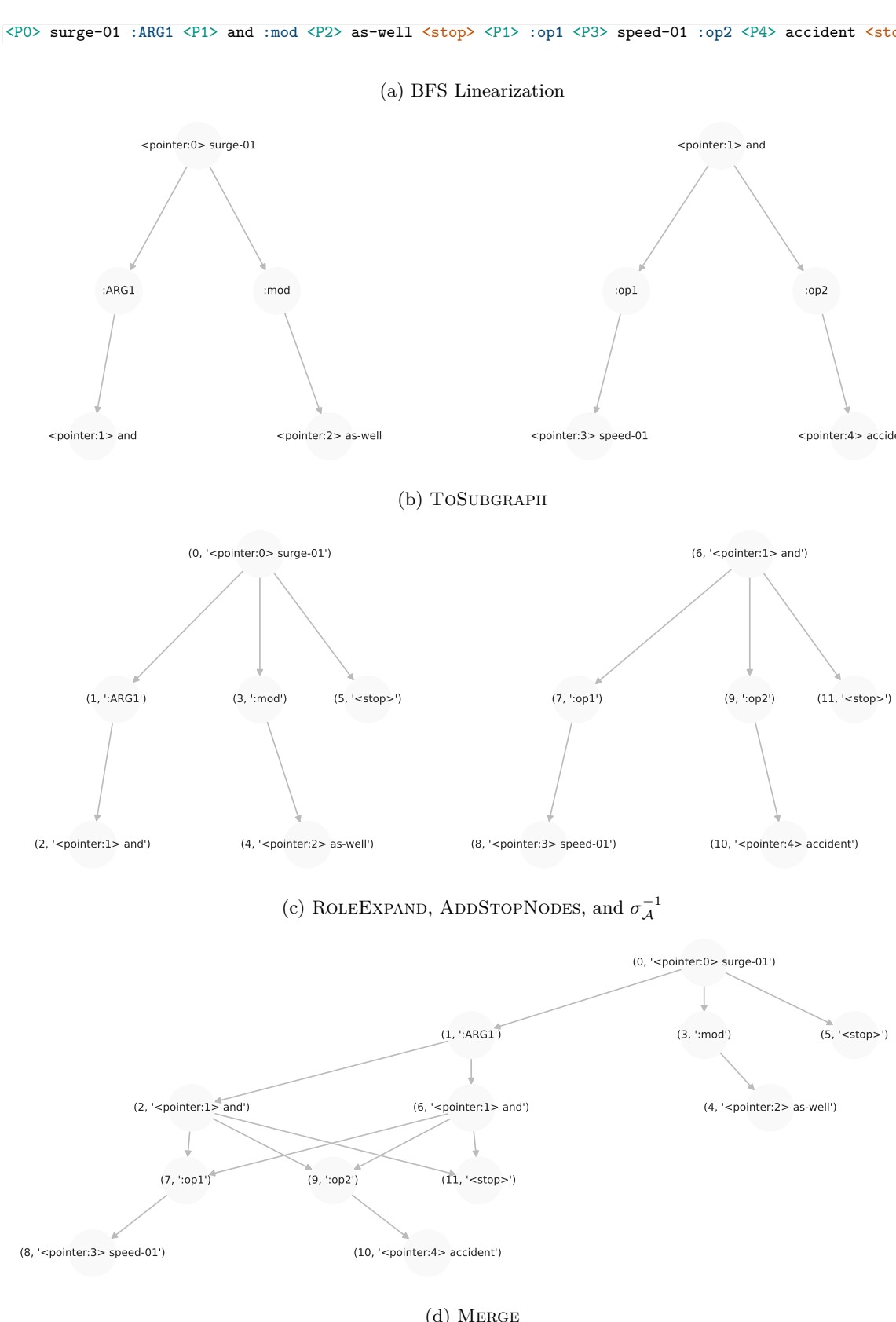

(b) ToSubgraph

(c) RoleExpand, AddStopNodes, and $\sigma_{\mathcal{A}}^{-1}$

(d) Merge

Figure 18: Overview of preprocessing steps for the AMR corresponding to the sentence: "Speeding and accidents have surged as well".

