# OpenReview forum: "SAFT: Structure-Aware Fine-Tuning of Large Language Models for AMR-to-Text Generation"
_TMLR — Decision pending for TMLR_

### Review · Reviewer_Qavh · 2026-03-30

**Summary Of Contributions:**

This paper introduces SAFT (Structure-Aware Fine-Tuning), a method for fine-tuning decoder-only LLMs on AMR-to-text generation that incorporates graph positional encodings derived from the magnetic Laplacian. The authors linearize AMR graphs using BFS, transform them into semantic-preserving graphs (SPGs), compute positional encodings from the magnetic Laplacian eigenvectors, and inject these encodings into the LLM's token embeddings via a lightweight MLP projection.

Key Strengths:
- Application of magnetic Laplacian positional encodings to LLM fine-tuning for structured inputs
- Architecture-agnostic approach requiring no modifications to pretrained LLMs
- Comprehensive evaluation across multiple model families (LLaMA, Qwen, Gemma) and scales (0.5B-3B parameters)
- Thoughtful analysis showing gains increase with structural complexity (graph depth and document length)
- Strong empirical results that are competitive with prior work on AMR 3.0 sentence-level generation
- Detailed geometric analysis of how AmrPEs modify the embedding space
- Thorough Appendix with implementation details, additional experiments, and error analysis

Key Weaknesses:
- Limited scope of evaluation (only AMR-to-text; generalization to other graph-to-text tasks)
- Evaluation limited to models ≤3B parameters; scaling behavior beyond this range is unclear
- Choice of evaluation metrics not well justified (see detailed comments below)
- Computational cost analysis could be more comprehensive (only preprocessing time reported, additional MLP projection during training and inference is not quantified)

**Additional Comments:**

This is a well-executed, thorough paper that makes solid contributions to structure-aware LLM adaptation. The writing is generally clear (though dense in places due to technical details), experiments are comprehensive, and the method is practical. Addressing the metric justification would strengthen the work.

**Audience:**

Yes

**Audience Explanation:**

This work addresses an important question: how to effectively adapt LLMs to structured inputs without architectural modifications. The findings are relevant to researchers who work on:
- Structured data-to-text: The method provides a practical alternative to GNN-based adapters or linearization-only approaches
- LLM adaptation: The demonstration that precomputed, parameter-free positional encodings can inject useful inductive bias is broadly applicable
- Graph learning: Relatively unexplored application of magnetic Laplacian to NLP/LLM tasks
- Semantic parsing: Strong results on a well-established benchmark (AMR 3.0)

**Broader Impact Concerns:**

No significant broader impact concerns. The work is methodological and focuses on a well-established NLP task.

**Claims And Evidence:**

Yes

**Claims Explanation:**

The paper makes three primary claims, all reasonably well-supported:
1. SAFT improves or matches standard fine-tuning across model families and scales (Table 1): The evidence is convincing. The authors evaluate 6 different models and consistently show improvements. The bootstrap analysis (Table 6, Figure 9) demonstrates these gains are robust to test-set variation.
2. Gains increase with structural complexity: The stratified analyses by graph depth (Figure 2) and document size (Figure 3) provide compelling evidence.
3. AmrPEs expand embedding space dimensionality orthogonally to pretrained representations (Figure 4): The geometric analysis is thorough and well-presented.

Minor concerns:
- The comparison to prior work (Table 1) mixes different architectures (encoder-decoder vs decoder-only), making cross-group interpretation difficult. The authors acknowledge this appropriately.
- Generalization beyond AMR-to-text is claimed conceptually but not demonstrated empirically.

**Requested Changes:**

Critical:
1. Justify metric choices (Section 4.1, Table 1-2). The paper evaluates AMR-to-text generation using machine translation metrics (BLEU, chrF, METEOR, BERTScore) without explaining why these are appropriate. AMR-to-text is not a standard translation task, and multiple valid realizations exist for a single AMR graph, which raises questions about the suitability of MT metrics. Please either:
    - Justify why MT metrics are suitable for this task (e.g., citing prior work or task characteristics)
    - Why the evaluation focuses primarily on lexical-based metrics, with only one neural metric included
    - Add AMR-specific if they exist
    - Acknowledge this as a limitation inherited from prior work
2. Address neural metric selection (Table 2). You use BERTScore but not more recent neural metrics like COMET (widely considered state-of-the-art for MT evaluation) or BLEURT. Please either:
    - Add COMET scores (or discuss its suitability) to Table 2 and key results
    - Explain why BERTScore was chosen over alternatives (computational cost? task fit? prior work?)

Nice-to-have:
1. Error analysis (Appendix G). The qualitative examples are helpful but somewhat cherry-picked despite random sampling. Consider adding categorization of error types (fluency vs semantic vs structural)
2. Generalization experiment (Section 6, Appendix C.6). You claim SAFT is "conceptually applicable" to other graph-to-text tasks but only evaluate AMR. Even one small-scale experiment on another dataset would significantly strengthen generalization claims.

---

> ### Author Response · Authors · 2026-05-25
>
> Thank you for the detailed review and for the positive comments on the experiments and geometric analysis. We address the points you raised below. The manuscript already includes several clarifications noted below; other items are still running and will be added to the revision when ready.
>
> ---
>
> ## Critical 1: Justify metric choices (Section 4.1; Tables 1 and 2)
>
> BLEU, chrF, METEOR, and BERTScore are the metrics **commonly reported in AMR-to-text work** (e.g., SPRING, BiBL, StructAdapt, and related lines). We use the same set of metrics so our numbers are **directly comparable** to the baselines in Table 1 and to prior literature, rather than introducing a new evaluation protocol.
>
> As you correctly say, AMR-to-text is not a literal translation task, and a single graph can admit many valid verbalizations. **Section 4.1 (Experimental Setup)** now states this explicitly. The main limitation of the aforementioned metrics (i.e., that they can under-reward acceptable paraphrases) is **shared with much of the prior work we compare against**; we inherit that convention for consistency rather than claiming these metrics are ideal for semantic fidelity. We therefore report several metrics and stress **trends that hold across them** (including depth-stratified analyses in the appendix, where SAFT gains grow with structural complexity on METEOR and BERTScore as well as on BLEU/chrF). Human evaluation and task-specific automatic metrics remain important future work.
>
> ## Critical 2: Neural metric selection (BERTScore vs COMET / BLEURT)
>
> We agree that learned evaluators help contextualize the results.
>
> We did **not** use **COMET** in our main evaluation because it is pretrained for **machine translation** and expects a **source sentence** in one language, a hypothesis, and a reference translation. In AMR-to-text the input is an **AMR graph** (or its linearization), not a source-language sentence, and COMET models are **not trained** for that input type. Supplying the graph or gold text as a stand-in “source” would not reflect what the metric was built for.
>
> For the submission we report **BERTScore** in **Appendix C (Extra Evaluation Metrics)** alongside BLEU, chrF, and METEOR. We chose BERTScore for **computational cost and ease of use** on our saved generations: it only needs hypothesis and reference (both English) and is straightforward to reproduce.
>
> We are **currently running BLEURT** on the same AMR 3.0 test outputs and will add scores to the appendix (for at least two backbones from Table 1) with a short note on task fit.
>
> ## Weakness: Computational cost (preprocessing vs MLP at train/infer)
>
> **Appendix C (Runtime)** already separates one-time graph preprocessing from the per-forward two-layer MLP projection and notes that Transformer attention and FFN cost is unchanged relative to FT. It reports average preprocessing time on AMR 3.0 and DocAMR and states that the MLP is small relative to 0.5B-3B backbones.
>
> For **training overhead**, we compared matched FT vs SAFT runs on **Qwen 1.5B** (same LoRA recipe; SAFT uses $k=16$, $q=0.25$). Training time on 2170 steps was **27164 s** (FT) vs **28189 s** (SAFT), i.e. **3.8%** more for SAFT on the same run configuration. The projection MLP and graph encodings therefore add only a small training-time cost at this scale, consistent with the appendix parameter counts.
>
> On the same runs, total **evaluation** wall time (on 64 examples) was **50.1 s** (FT) vs **52.1 s** (SAFT), again only a few percent higher for SAFT.
>
> ## Weaknesses noted in summary (scope and scale)
>
> - **Only AMR-to-text empirically:** **Appendix C (Potential applicability beyond AMR)** keeps broader applicability conceptual; we do not claim cross-domain experiments in this paper.
> - **Models at most 3B:** Results use LoRA on open models from 0.5B to 3B. We are **running larger open backbones** on the same AMR 3.0 / DocAMR protocol where possible. These jobs need much more GPU time than the 3B grid, so we may not have full numbers before the first rebuttal post; we will add stable results to the revision and note any late updates in this thread.
>
> ## Nice-to-have 1: Error analysis (Appendix G)
>
> Appendix G uses random sampling. If time allows, we will add a short breakdown (fluency vs semantic vs structural) on a labeled subset. We will also clarify that examples illustrate typical failures, not cherry-picked successes.
>
> ## Nice-to-have 2: Generalization beyond AMR
>
> Despite our empirical scope being focused on AMR 3.0 and DocAMR, we agree with you that applicability to other settings is of strong interest for future work.
> We are currently adapting our pipeline to WebNLG, but this requires some engineering adaptations, such as data pipelines, and long fine-tuning for both FT and SAFT. We will report results on this additional setting if time permits.
>
>
> We will post a short follow-up here if new numbers land after this response.
>
> ---
>
> We thank you again for the review.

---

### Review · Reviewer_pUkB · 2026-04-07

**Summary Of Contributions:**

The paper proposes SAFT, a structure-aware fine-tuning method that augments decoder-only LLMs with graph positional encodings derived from the magnetic Laplacian of AMR graphs, injected into token embeddings via a small MLP without modifying the LLM architecture. The authors show consistent gains over standard fine-tuning across several 0.5B–3B open LLMs on AMR 3.0, with benefits that grow with graph depth and on document-level AMRs. They further analyze the representational geometry induced by the encodings, arguing that SAFT injects an orthogonal, structured signal that expands the intrinsic dimensionality of token representations.

**Audience:**

Yes

**Audience Explanation:**

This paper addresses an important open question: how to add structural inductive bias to pretrained LLMs without architectural changes. Empirical results suggest the method is robust, inexpensive, and scales with structural complexity, and useful for structure-to-text tasks where faithfulness is critical.

**Broader Impact Concerns:**

Potential risks may include over-reliance on spectral signals for noisy graphs and the possibility of injecting spurious biases if SPG construction is imperfect.

**Claims And Evidence:**

Yes

**Claims Explanation:**

The proposed method is straightforward and well organized for presentation. The results are promising and are tested across different LLM families. And the implementation details are clear to support the method, facilitating the reproducibility.

**Requested Changes:**

1. The choice and sensitivity of key spectral hyperparameters are under-specified. It is unclear how robust performance is to these settings.
2. The zero/few-shot GPT-4o comparisons underscore the need for fine-tuning, but are not definitive. Prompting quality and number of shots can matter greatly.
3. The method is clear and easy to follow. But the ablation analysis of each component is lacking.

---

> ### Author Response · Authors · 2026-05-25
>
> Thank you for the thoughtful review. We respond below to each requested change. Where the manuscript already states our position, we point to the relevant section; otherwise we report new analyses or note what we are still running.
>
> ---
>
> ## R1: Spectral hyperparameters are under-specified
>
> Thank you for this feedback. We show sensitivity to spectral hyperparameters on **LLaMA 3.2 (3B)**, using the same training protocol as the main results.
>
> **Number of eigenvectors** $k \in \{8, 16, 30, 48\}$: BLEU rises from **52.4** to **55.0** without a sharp drop at intermediate values. Table 1 uses **$k=30$** (54.2 BLEU on this backbone); the sweep shows the result is not tied to a single lucky $k$.
>
> | Eigenvectors $k$ | 8 | 16 | 30 | 48 |
> |:--:|:--:|:--:|:--:|:--:|
> | BLEU $\uparrow$ | 52.4 | 53.6 | 54.2 | 55.0 |
>
> **Laplacian type:** FT **53.5**; SAFT with a combinatorial (undirected) Laplacian **53.9**; SAFT with the magnetic Laplacian **54.2**.
>
> | Laplacian | BLEU $\uparrow$ |
> |:--|:--:|
> | FT (no Laplacian) | 53.5 |
> | SAFT (combinatorial) | 53.9 |
> | SAFT (magnetic) | 54.2 |
>
> The directed Laplacian helps beyond an undirected one. These numbers are in **Appendix C** (Sensitivity and component ablations). Appendix D already notes that $k$ and the magnetic phase $q$ may need tuning; the implementation appendix cross-references the sensitivity table.
>
> We are running a small sweep over $q$ on the same backbone. A full grid over every spectral knob and every model family is beyond what we can finish in the rebuttal period.
>
> ## R2: GPT-4o zero/few-shot comparison is not definitive
>
> We agree. Prompt format and shot count can change closed-model scores a lot, and our table should not be read as the best achievable prompting result.
>
> In **Section 4.4**, we evaluate GPT-4o(-mini) with a **fixed prompt and seven in-context AMR-text pairs** (template in the appendix). We did not push to a larger shot count via the API: each demonstration pairs a full AMR linearization with its target sentence, so the prompt grows quickly and additional examples risk hitting **context-length limits**. Seven pairs were the practical ceiling in our runs. The text states that our comparison to fine-tuning is limited to that setup. In our runs, going from zero-shot to few-shot helps only slightly, and both stay well below a fine-tuned 3B open model. We use this to motivate fine-tuning and structural encodings in our setting, **not** to claim that prompting cannot work with other designs.
>
> ## R3: Component ablations are lacking
>
> Table 1 already compares **FT** and **SAFT** under the same LoRA setup on each backbone: FT uses no graph positional encodings, SAFT adds them (with the projection MLP). That is the main with/without-SAFT comparison, reported for six models. On **LLaMA 3.2 (3B)** and AMR 3.0, FT reaches **53.5** BLEU and SAFT **54.2** BLEU.
>
> On the same 3B setup we also ran two smaller checks:
>
> - **Laplacian** (see R1): combinatorial **53.9**, magnetic **54.2**. Dropping directionality lowers score, but SAFT still beats FT.
> - **MLP depth:** one layer **40.7**, two layers **54.2**. The two-layer projector used in SAFT is important.
>
>     | MLP depth | BLEU $\uparrow$ |
>     |:--|:--:|
>     | 1 layer | 40.7 |
>     | 2 layers (default) | 54.2 |
>
> They are reported in **Appendix C** alongside Table 1 in the main text.
>
> ## R4: Broader impact (noisy graphs and imperfect SPG construction)
>
> We share both concerns. SAFT builds on BFS linearization and SPG conversion (Section 3). If the AMR or SPG is wrong, the Laplacian encodings reflect that error like any other input feature. In noisy or low-trust graph settings, outputs should be checked accordingly; SAFT is not a substitute for graph validation.
>
> We added a paragraph to **Appendix D (Limitations)** on dependence on graph quality, imperfect SPG construction, and simple mitigations (e.g., validating parses, or falling back to FT when structure is unreliable). This is clarification in the text rather than a new experimental study on corrupted or adversarial graphs.
>
> ## Still in progress
>
> - Bounded sweep over magnetic phase $q$ on LLaMA 3.2 (3B). We will post an update in this thread if it finishes before recommendations.
>
> ---
>
> Thank you again for the useful comments on robustness and ablations.

---

### Review · Reviewer_LKHN · 2026-05-12

**Summary Of Contributions:**

This paper proposes SAFT, a lightweight framework for incorporating graph structural information into transformer-based AMR-to-text generation through positional encodings derived from the magnetic Laplacian. Unlike graph-to-sequence architectures that modify the backbone architecture, SAFT preserves the standard Transformer design and injects graph topology and directionality information through spectral positional embeddings.

The paper evaluates the method on AMR 3.0 and DocAMR benchmarks using several open-source LLM backbones and compares SAFT against standard fine-tuning baselines. The approach is conceptually clean, lightweight, and easy to integrate into existing LLM fine-tuning pipelines.

Strengths of the paper include:

1. A technically elegant use of magnetic Laplacian positional encoding for directed graph structures.
2. Clear motivation for improving graph structural awareness in transformer models.

Weaknesses include:

1. Unclear practical significance of AMR-to-text generation in the current LLM era.
2. Limited empirical improvements over standard fine-tuning.
3. Lack of comparison against graph-to-sequence methods.
4. Limited experimental setting in terms of datasets, model sizes, and fine-tuning settings.

**Additional Comments:**

No.

**Audience:**

Yes

**Audience Explanation:**

Yes. The paper explores an interesting direction for incorporating graph structural information into transformer-based LLM fine-tuning without modifying the backbone architecture. The idea of injecting graph topology through positional encodings is technically elegant and may generalize beyond AMR to broader graph-structured domains.

In particular, researchers working on graph representation learning, graph-aware LLMs, semantic parsing, or lightweight adaptation methods may find the proposed framework interesting. The paper may also motivate future work on applying spectral positional encodings to other graph modalities such as molecular graphs or knowledge graphs, although not tested in this paper.

That said, the practical importance of AMR itself in the current LLM era is less clear than before, which somewhat limits the broader impact of the current experimental scope.

**Broader Impact Concerns:**

No concern.

**Claims And Evidence:**

Yes

**Claims Explanation:**

The empirical results generally support the claim that incorporating graph structural information through positional encoding can improve AMR-to-text generation performance over standard fine-tuning baselines. In particular, the improvements on DocAMR suggest that the method may become more useful as graph structures become more complex.

However, the current evidence is still somewhat limited for fully establishing the general effectiveness of the proposed approach.

First, the observed gains over standard fine-tuning are relatively marginal on AMR 3.0, often around 1–2 points. Although better for DocAMR, it remains unclear under what structural conditions SAFT provides substantial benefits versus marginal improvements.

Second, the experiments focus primarily on LoRA and relatively small models. The paper does not investigate whether the advantage persists under full-parameter fine-tuning or with larger frontier models.

Third, the paper lacks comparisons against graph-to-sequence approaches, which makes it difficult to properly position SAFT within the broader graph generation literature and evaluate its practical trade-offs.

Overall, the evidence supports the technical validity of the method, but additional experiments would substantially strengthen the paper’s claims regarding generality and practical impact.

**Requested Changes:**

Major changes

1. Expand the empirical evaluation to other graph datasets.
The importance of AMR in the era of large language models is highly questionable. Historically, AMR has mainly served as a tool for helping language models understand semantics. However, with the rise of modern LLMs, semantic understanding no longer heavily depends on AMR representations. As a result, improving AMR-to-text generation may not carry substantial significance today. That said, since the core idea of SAFT could potentially generalize to different types of graph-structured data, the paper would benefit from broadening its scope toward a more general framework for handling graph data using positional encodings, with AMR presented as only one application scenario. When select datasets, the authors should place greater emphasis on practical relevance, for example, molecular graphs or knowledge graphs, rather than AMR.

2. Evaluate on more datasets and clarify under what conditions SAFT provides meaningful gains.
The advantage of SAFT over standard fine-tuning (FT) appears rather limited. As shown in Table 1, the gains are often only around 1–2, which is relatively marginal. Although Figure 3 demonstrates larger improvements on DocAMR, it remains unclear under what kinds of benchmarks or structural conditions SAFT yields substantial gains versus negligible ones. Combined with the weak performance of GPT-4o on this task, one possible interpretation is that LLMs perform poorly simply because their pretraining data consists mostly of natural language and contains very little graph-structured data. Once some fine-tuning is performed on such graph data, performance quickly improves regardless of the fine-tuning strategy used. If the authors could include a broader set of datasets and consistently demonstrate that SAFT outperforms FT across most of them, this would provide much stronger evidence for the effectiveness of the proposed method.

3. Evaluate more training/model settings.
The experimental setting explored in the paper is limited. The experiments mainly compare standard fine-tuning + LoRA against SAFT + LoRA, and primarily focus on relatively small models. It is possible that under full-parameter fine-tuning, or with larger models, the gap between SAFT and FT would become much smaller or even negligible.

4. Include comparisons against graph-to-sequence (G2S) methods.
The paper does not include comparisons against graph-to-sequence (G2S) methods on the corresponding datasets. Such comparisons are important for properly positioning the contribution of SAFT. If G2S methods substantially outperform FT, while SAFT achieves performance much closer to G2S than to standard FT, then SAFT could be viewed as an excellent engineering trade-off: significantly improving performance without introducing specialized graph architectures. However, if SAFT remains much closer to the FT baseline, then the practical value of the method becomes considerably more limited. From an engineering perspective, one could simply use standard FT for simplicity and convenience, or adopt specialized graph architectures when high accuracy is required. In this case, SAFT introduces additional engineering complexity while providing only marginal improvements over FT, resulting in a relatively weak cost-benefit trade-off.

Minor changes:

1. Present each dataset using a unified “one table + one figure” format for clarity and consistency.
It would improve clarity if the experimental presentation for each dataset followed a unified “one table + one figure” format. The table could follow the style of Table 1, but should include: GPT-4o zero-shot, GPT-4o few-shot, as well as zero-shot, few-shot, FT, and SAFT results for each small model. In addition, besides GPT-4o, the paper should include a few more recent frontier models, such as Gemini 3, GPT-5.4, or Claude Sonnet 4.6. Separate tables like Table 2 for large models may then become unnecessary. The figure could follow the style of Figure 3, which more clearly illustrates how the performance gap between FT and SAFT evolves as graph complexity increases.

2. Use more recent open-source models
The models used in the paper, including Qwen2.5, Llama 3.2, and Gemma, are already somewhat outdated. More recent model versions should be included in the experiments.

---

> ### Author Response · Authors · 2026-05-25
>
> Thank you for the careful review and for noting both the spectral-encoding idea and the limits of our current empirical scope. We respond to each requested change below.
>
> ---
>
> ## Major 1: AMR role in the LLM era and SAFT beyond AMR
>
> **1. AMR’s role today.** We agree that AMR is no longer as central to mainstream NLP as when these benchmarks were established: modern LMs already encode much semantic knowledge from text, so improving AMR-to-text is not the same as claiming AMR is essential for production semantic pipelines. We do not make that claim.
>
> **2. Why we still study AMR-to-text in this paper.** Even with strong LMs, **AMR-to-text remains difficult** without task-specific training (Section 4.4). The task couples structured input with open-ended generation and is a **controlled testbed** for structure-aware fine-tuning of decoder-only models. Our empirical focus is methodological, rather than a statement that AMR is today’s most important application.
>
> **3. SAFT beyond AMR (conceptually).** The core method is not AMR-specific: it requires a directed graph and node–token alignment. **Appendix C (Potential applicability beyond AMR)** discusses extension in principle. AMR is the application we implement and evaluate thoroughly in this submission.
>
> **4. Other graph datasets in this revision.** Experiments on **molecular graphs** or **knowledge graphs** at scale would need new data pipelines and training runs we have not built for this paper. A closer graph-to-text extension on our existing codebase is **WebNLG** (see our reply to Reviewer Qavh); we will report FT/SAFT there if runs finish in time. The domains you highlight remain important **follow-up** beyond the AMR-focused scope of the current revision.
>
> ## Major 2: More datasets; when SAFT gives meaningful gains vs marginal ones
>
> **1. AMR coverage is broader than a single corpus.** Table 1 reports aggregate scores on **AMR 3.0** (LDC2020T02), which already merges several **source subdatasets and genres** (newswire, weblogs, discussion forums, broadcast conversation, fiction; **Appendix E**). We also evaluate **DocAMR**, a document-level regime built from the same AMR 3.0 test sentences with added cross-sentence structure. So our empirical scope is AMR-focused, but not a single homogeneous domain.
>
> **2. Gains are not uniform—they grow with structural complexity.** Aggregate BLEU on AMR 3.0 is often modest (about 1–2 points in Table 1), while DocAMR shows larger separation. The paper explains **when** SAFT helps:
>
> - **AMR 3.0 (Figure 2):** FT–SAFT gaps increase with graph depth.
> - **DocAMR (Figure 3):** the gap widens as document-level complexity grows.
>
> Our claim is therefore not “SAFT wins everywhere on AMR,” but that **structural complexity predicts larger benefits**. We will make that reading easier to find in the main text if needed.
>
> **3. Fine-tuning on graph-linearized data vs. SAFT specifically.** One interpretation is that LLMs struggle on AMR largely because pretraining is text-heavy, and that performance improves substantially once models are fine-tuned on graph data, with limited remaining benefit from the fine-tuning *strategy* (SAFT vs. FT). On **shallow** AMR 3.0 instances, where Table 1 gaps are small, that interpretation is plausible. It is less consistent with our **depth-stratified** and **DocAMR** results, where FT–SAFT separation **grows** as structure becomes harder. We will state this distinction more explicitly in the text.
>
> **4. Scope of claims in this revision.** We do **not** claim SAFT beats FT on most datasets overall. The submitted paper rests on AMR 3.0 and DocAMR. As noted in our reply to Reviewer Qavh, we are adapting to **WebNLG** and will report FT/SAFT there if runs finish in time; until then, those numbers are not part of the revision’s empirical claims.
>
> ## Major 3: Full-parameter fine-tuning and larger models
>
> All reported FT/SAFT pairs use **LoRA** on open models up to about 3B. We do not report full-parameter fine-tuning or proprietary frontier APIs in this revision.
>
> We are **running larger open-weight backbones (~7B)** under the same protocol where resources allow (see also our reply on scale in the Qavh thread). Full fine-tuning and broad API sweeps remain follow-up work.

---

> ### Author Response · Authors · 2026-05-25
>
> ## Major 4: Comparisons against graph-to-sequence (G2S) methods
>
> **1. What Table 1 already reports.** **Section 4.2** and Table 1 include strong prior **graph-based and seq2seq** AMR-to-text systems on the same AMR 3.0 test set: **SPRING** and **BiBL** (linearized seq2seq) and **StructAdapt** (graph2seq), alongside our decoder-only **FT** and **SAFT** lines (LoRA on modern LLMs). So the paper already situates SAFT next to graph-native baselines, not only next to FT.
>
> **2. How to read that comparison.** These rows are **not** matched experiments (different backbones, pretraining, and training recipes). We state that limitation explicitly and do not treat Table 1 as a controlled G2S ablation. Still, on aggregate AMR 3.0 BLEU without extra silver data, our best **SAFT** entries (e.g., 54.2 on LLaMA 3.2 3B) are **above** the listed graph-native systems (StructAdapt 48.0; BiBL 47.4; SPRING 44.9), while **FT** is already competitive. That pattern is consistent with an attractive engineering trade-off—gains from structure-aware fine-tuning without adding a separate graph encoder—though cross-system numbers should be interpreted with caution.
>
> **3. SAFT vs. FT within our setup.** On every backbone in Table 1, **SAFT** improves over **FT** under the same LoRA protocol. That is the controlled comparison for our method; it is separate from ranking against historical G2S systems.
>
> **4. What we do not add in this rebuttal.** A **matched** G2S baseline on every slice (e.g., DocAMR, same compute budget and backbone class) would sharpen the story further; we do not have new G2S training runs in this period. DocAMR in the paper compares **FT** vs. **SAFT** only. Extending matched G2S comparisons there remains follow-up work.
>
> ## Minor 1: Unified table + figure per dataset; more frontier closed models
>
> Thank you for the suggestion. We understand this as a **presentation** improvement: for each evaluation setting, one summary **table** (all relevant systems) and one **figure** in the style of Figure 3 (how the FT–SAFT gap evolves with structural complexity). In our paper that naturally maps to **(i) sentence-level AMR 3.0** and **(ii) DocAMR**, rather than many separate corpora.
>
> We already have much of this material—Table 1 and Figure 2 (depth) for AMR 3.0, Figure 3 for DocAMR, and Section 4.4 for GPT-4o(-mini) / GPT-4o zero- and few-shot—but the pieces are split across sections. We will **try to reorganize** the revision toward a clearer per-setting layout (e.g., bringing prompting baselines closer to the main AMR 3.0 comparison where space allows, and tightening cross-references to Figures 2–3). Our open models are reported as **FT** vs **SAFT** after task fine-tuning; we do not have zero- or few-shot rows for every small backbone.
>
> A broader sweep of additional proprietary APIs (Gemini, Claude, etc.) is unlikely in this revision window; the GPT-4o study remains under the fixed prompt in Section 4.4. We will prioritize the table/figure structure for the results we already report.
>
> ## Minor 2: More recent open-source model families
>
> We agree that model families and versions move quickly. We are **currently prioritizing larger open-weight backbones** (~7B, same FT/SAFT protocol; see Major 3). If time permits before the revision deadline, we will add runs on **more recent open-source families**; otherwise we treat a full refresh of the model grid as follow-up work.
>
> ---
>
> Thank you again for the constructive feedback.

---

### Decision · Action_Editor_wtzf · 2026-07-05

**Recommendation:** Accept with minor revision

**Additional Comments:**

The authors are encouraged to integrate material from the rebuttal, as well as scope the claims more carefully as described above. If those changes are made, I believe the paper can be accepted for TMLR.

**Audience:**

Yes

**Audience Explanation:**

This is a bit trickier. As the paper is currently restricted to AMR parsing, the relevance to broader semantic graphs and structured decoding is limited. Many practitioners using other formalisms or other structures probably won't see much value here. As a result, I think the target audience for this paper is small. But if the question is whether there is some target audience, the answer is yes.

**Claims And Evidence:**

Yes

**Claims Explanation:**

This paper claims to introduce "a structure-aware fine-tuning method that augments LLMs with graph positional encodings derived from the magnetic Laplacian of the input graph." The reviewers agree that this
 method is straightforward and interesting and the evaluation on AMR is mostly sound, although limited to relatively small-scale LLMs.

I do think the claims should be edited somewhat. In particular, the authors should avoid overclaiming that their method is generalizable beyond AMR. It is okay to claim that it is *conceptually* generalizab
le, but any claims about experiments should be limited to AMR. For example, the third contribution:

> Empirical evidence that SAFT consistently improves or matches standard fine-tuning across model families and scales(Table1) [...]

should make it clear that this is for the AMR task.